# Not All Prefills Are Equal:
# PPD Disaggregation for Multi-turn LLM Serving

**Zongze Li** [1]  **Jingyu Liu** [1]  **Zhen Xu** [1]  **Yineng Zhang** [2]  **Tahseen Rabbani** [1]  **Ce Zhang** [1]

## Abstract

Prefill-Decode (PD) disaggregation has become the standard architecture for modern LLM inference engines, which alleviates the interference of two distinctive workloads. With the growing demand for multi-turn interactions in chatbots and agentic systems, we re-examined PD in this case and found two fundamental inefficiencies: (1) every turn requires prefilling the new prompt and response from the last turn, and (2) repeated KV transfers between prefill and decode nodes saturate the bandwidth, leading to high latency and even service degradation. Our key insight is that not all prefill operations are equally disruptive: *append-prefill*, which processes only the new input tokens while reusing cached KV states, incurs an order-of-magnitude smaller decoding slowdown than full prefill. This motivates routing append-prefill to decode nodes locally. However, through comprehensive analysis, we show that *no single fixed routing strategy satisfies all Service Level Objectives (SLOs) simultaneously*. Based on this insight, we propose **P**refill **P**refill-capable **D**ecode (PPD) disaggregation, a dynamic routing system that decides when to process Turn 2+ requests locally on decode nodes using cached KV states. PPD adapts to varying SLOs via configurable weights and seamlessly integrates with traditional PD deployments. With extensive evaluations, we show that PPD reduces Turn 2+ time-to-first-token (TTFT) by ∼68% while maintaining competitive time-per-output-token (TPOT), effectively alleviating KV transfer congestion under high load. PPD provides a flexible and efficient paradigm for multi-turn LLM serving.

[1]University of Chicago, USA [2]Independent Researcher. Correspondence to: Zongze Li <zongzel@uchicago.edu>.

*Proceedings of the 43rd International Conference on Machine Learning*, Seoul, South Korea. PMLR 306, 2026. Copyright 2026 by the author(s).

## 1. Introduction

Prefill-decode (PD) disaggregation has become the standard architecture for LLM inference (Zhong et al., 2024; Patel et al., 2025; DeepSeek-AI et al., 2025; Sun et al., 2024), which places compute-intensive prompt processing (prefill) and memory-bound token generation (decode) separately on dedicated GPU pools. Despite being effective for independent single-turn queries, PD exhibits critical inefficiencies under *multi-turn conversations*, the dominant usage pattern in chatbots and agentic systems (Duan et al., 2023). When a typical PD handles a new-turn query, PD routes the prompt to the prefill node along with the response string from the previous turn. A subtle but consequential property of canonical PD designs is that KV transfer is strictly P→D: P operates as a producer, D as a consumer, with no reverse path. Even though the response token KV from the previous turn already resides on D, it is inaccessible to P; the prefill node must therefore recompute the KV cache for the entire conversation history (prior responses plus the new prompt) before transferring it back to D. Recent measurements on real chat workloads find that this recomputation accounts for up to 99% of multi-turn prefill cost (Gao et al., 2024). We re-examine PD under the case where multi-turn conversations are prevalent, and find that the standard PD strategy causes time-to-first-token (TTFT) for Turn 2+ (the second turn and beyond) to remain high even with shorter inputs, and that these repeated KV transfers saturate network bandwidth (Section 6).

Our key insight is that the behavior of prefill operations differs sharply between the first and later turns, motivating a specialized strategy. In Figure 2, we demonstrate that *not all prefill operations are equally disruptive for decoding*: full prefill (a new prompt without cached context) causes an order of magnitude greater decode slowdown than *append-prefill* (only the new input tokens, reusing cached KV). This order-of-magnitude gap indicates that running append-prefill (AP) on decode nodes locally can efficiently handle Turn 2+ prompts with minimal interference. While a concurrent work (He et al., 2026) shares the high-level intuition of routing incremental prefills to decode nodes, our work uniquely grounds this approach in a rigorous micro-architectural interference analysis and formalizes the routing

decision as an optimization problem.

On the one hand, routing AP operations to prefill nodes can maintain a high TPOT; on the other hand, routing to the decode nodes can improve overall throughput. The central question is therefore *should we route the AP operations to the prefill or decode nodes*? We formalize this problem with a common framework in which PD is a special case that always routes AP to the prefill nodes. To understand the trade-offs (see Section 4), we routed a different fixed percentage of AP to decode nodes for each strategy, and we found *no single fixed strategy meets all Service Level Objectives (SLOs)* (see Table 2).

To this end, we introduce **P**refill **P**refill-capable **D**ecode (PPD) disaggregation (Section 5), which proposes to dynamically route AP operations to the decode nodes based on the current workload estimate, user-specified SLO, and initial node assignment. PPD optimizes the disaggregated serving objective with a simple yet effective algorithm using precomputed offline statistics, and can always recover the default PD configuration when it needs to. In the extreme case, PPD can route all AP operations to the decode nodes, where the locally cached KV states can be fully reused without extra recomputation or KV transfers.

On both synthetic and real datasets, we evaluate the effectiveness of our proposed PPD framework and show that PPD achieves the best Pareto frontier compared to fixed routing or the default PD (see Figure 1). We also conduct detailed ablation studies to justify our design choices and analyze complex serving scenarios with varying requirements. We summarize our main contributions as follows.

**(i)** We identify the inefficiencies of PD in multi-turn conversations (Section 4.1): running AP on the prefill node incurs extra recomputation and saturates network bandwidth via frequent KV cache transfers.

**(ii)** We find that full prefill and AP differ by an order of magnitude in decode interference (48% vs. 2% TPOT slowdown at batch size 200), motivating selective AP-to-D routing.

**(iii)** We formalize multi-turn inference serving as an optimization problem in which traditional PD is the special case $x \equiv 0$, and show that no single fixed strategy dominates (Section 4.4).

**(iv)** We propose PPD, a dynamic routing system that selects $x$ per request from current workload, operator weights, and initial node assignment.

**(v)** Detailed evaluations (Section 6) show PPD outperforms standard PD and fixed strategies, achieving 48–73% Turn 2+ TTFT reduction on synthetic sweeps while maintaining competitive TPOT.

## 2. Background

### 2.1. LLM Inference and Scaling

LLM inference proceeds in two phases. *Prefill* processes the input prompt in parallel, producing the first output token and populating the KV cache; it is compute-bound with $O(n^2)$ attention complexity for $n$ tokens (Liu et al., 2025a; Shi et al., 2024), though FlashAttention (Dao et al., 2022; Shah et al., 2024) reduces memory complexity from quadratic to linear. *Decode* then generates tokens autoregressively, loading the full model weights for each token; it is memory-bandwidth-bound (Yu et al., 2022; Kwon et al., 2023). Grouped Query Attention (GQA) (Ainslie et al., 2023) reduces the KV cache memory footprint while maintaining model quality. The KV cache (key-value states from prior tokens) grows linearly with context and, alongside model weights, dominates GPU memory.

The most straightforward approach to scaling LLM serving is *replication*: deploying identical model instances across GPUs, each handling requests independently. However, colocating prefill and decode on the same GPU leads to *prefill-decode interference* (Zhong et al., 2024): long-running prefill computations block decode iterations, causing unpredictable latency spikes for ongoing generation. Chunked-prefill techniques (Agrawal et al., 2024) mitigate this by splitting prefill into smaller chunks that interleave with decode; DeepSpeed-FastGen (Holmes et al., 2024) further introduces Dynamic SplitFuse for efficient token composition. However, these techniques cannot fully eliminate interference when prefill workloads are heavy (we quantify this gap in Section 4.1).

### 2.2. Prefill-Decode Disaggregation

Prefill-decode (PD) disaggregation mitigates interference by physically separating prefill and decode onto distinct GPU pools (Zhong et al., 2024; Patel et al., 2025). Prefill nodes (P) process incoming prompts and transfer the resulting KV cache to decode nodes (D) over the network; D then performs autoregressive generation without interruption. This architecture eliminates interference, enables independent scaling of P and D resources, and permits hardware heterogeneity (Patel et al., 2025).

PD disaggregation has rapidly become the industry standard. It is supported by all major serving frameworks (vLLM (Kwon et al., 2023), SGLang (Zheng et al., 2024), TensorRT-LLM, LMDeploy (Contributors, 2023), and NVIDIA Dynamo (NVIDIA, 2025)) and is deployed at production scale by providers such as DeepSeek (DeepSeek-AI et al., 2025). However, disaggregation introduces a fundamental cost: every request requires transferring the full KV cache over the network. For Llama-3.1-8B with a 2K-token context, each transfer is ~256 MB. Recent work ex-

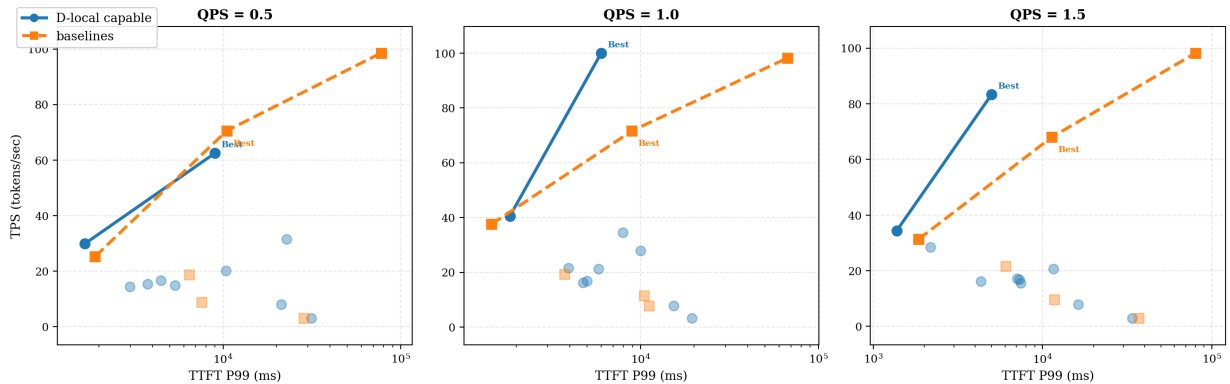

*Figure 1.* **99th-percentile (P99) TTFT vs. Tokens-per-Second (TPS) Pareto frontiers** under a long-context workload (10k input, 100 output tokens, 5 turns) at three load levels. Higher TPS and lower TTFT are better (upper-left is ideal). *Baselines* (orange): PD and Replica configurations where Turn 2+ requests always require P-node processing. *D-local capable* (blue): configurations allowing decode nodes to process Turn 2+ locally via append-prefill. The "Best" annotation marks the configuration selected by our dynamic PPD routing system, validating correct trade-off optimization. See Figure 9 in the Appendix for small-context results showing consistent trends across different workloads.

plores alternative disaggregation strategies: DuetServe (Gao et al., 2025) achieves disaggregation-level isolation within a single GPU through adaptive SM partitioning; Nexus (Shi et al., 2025) proactively reallocates GPU resources between phases; and TaiChi (Wang et al., 2025) unifies aggregation and disaggregation, showing that the optimal strategy depends on SLO constraints.

**One-way KV Transfer Protocol.** A defining architectural property of PD disaggregation is the directionality of its KV transfer protocol: P nodes act as KV *producers* and D nodes as *consumers*, with no reverse channel from D back to P. This producer/consumer contract is preserved across all major production engines (Zhong et al., 2024; Kwon et al., 2023; Zheng et al., 2024). A direct consequence in multi-turn serving is that any KV state computed on D, including all decoded responses, is unreachable from P, so each new turn re-traverses the full P→D pipeline (Section 1). External KV cache layers (Qin et al., 2025; Hu et al., 2024a; Gao et al., 2024) address this by adding a *separate* storage tier outside the disaggregation topology, rather than altering the producer/consumer contract itself.

### 2.3. Multi-turn Serving and KV Cache Reuse

Real-world LLM deployments are dominated by multi-turn conversations: chatbots and agentic systems typically involve multiple turns per session (Jeong & Ahn, 2026). Under PD disaggregation, each new turn is sent to a P node, which re-computes the entire conversation's KV cache (including prior outputs), then transfers it to a decode node for autoregressive generation. Historical tokens dominate input length in later turns (Gao et al., 2024), making this recomputation the dominant prefill cost in multi-turn workloads.

**Existing Approaches.** A growing line of work addresses this via *external KV cache stores*: CachedAttention (Gao et al., 2024) uses hierarchical caching; Mooncake (Qin et al., 2025) provides cluster-wide distributed KV stores; MemServe (Hu et al., 2024a) unifies context caching with disaggregation; and LMCache (Liu et al., 2025b) offers a modular KV cache layer. Prefix caching in SGLang (Zheng et al., 2024) and vLLM (Kwon et al., 2023) enables KV reuse for repeated prefixes. Complementary approaches include selective recomputation (Yao et al., 2025), CPU offloading (Chen et al., 2024; Sun et al., 2025), KV compression (Li et al., 2024; Liu & Zhang, 2025), streaming (Liu et al., 2024), prefix-aware attention (Ye et al., 2024), and compressive memory (Munkhdalai et al., 2024).

These approaches share a common strategy: storing KV caches externally or requiring requests to return to the same instance. We take a complementary approach: adjusting *routing* so follow-up turns execute directly on the decode GPU that already holds the KV cache. A concurrent work, AMPD (He et al., 2026), also explores a routing-based approach to mitigate multi-turn inefficiencies, albeit using real-time queue states rather than our offline optimization framework. We discuss the relationship with distributed KV stores and concurrent routing systems in Section 7.

## 3. Dynamic Routing of Append-Prefill Operations as an Optimization Problem

The central question is *where* to compute **A**ppend-**P**refill (AP) operations, on prefill or decode nodes, in disaggregated serving. We use $x$ in two related senses: a hardware-level fraction $x \in [0, 1]$ of AP routed to D (uniform across requests; Section 4), and a per-request binary decision $x \in \{0, 1\}$ (route through P or process locally on D; Section 5).

Section 4 sweeps five static fractions $x \in \{0, \frac{1}{3}, \frac{1}{2}, \frac{2}{3}, 1\}$ and three P:D node assignments $\pi$ (1P_3D, 2P_2D, 3P_1D), under workloads $\psi$ that span decode-heavy, balanced, and prefill-heavy profiles. We focus on pure P/D configurations; hybrid R+P/D configurations generally underperform (Section 4.2). No single static $x$ dominates TTFT, TPOT, and throughput simultaneously, which motivates a per-request policy.

Given $\pi$ and operator-specified weights $\mathbf{w} = (w_{\text{ttft}}, w_{\text{tpot}})$, we score the benefit of local processing ($x{=}1$) over the PD path ($x{=}0$) for workload $\psi$ as

$$S(\psi; \pi, \mathbf{w}) \;=\; w_{\text{ttft}}\, \Delta_{\text{ttft}} \;-\; w_{\text{tpot}}\, \Delta_{\text{tpot}}, \qquad (1)$$

where $\Delta_{\text{ttft}}$ is the relative TTFT improvement and $\Delta_{\text{tpot}}$ the relative TPOT degradation when $\psi$ is processed locally rather than through P. Each Turn 2+ request is routed locally when $S > 0$ and through P otherwise; Turn 1 always uses the PD path since no cached KV exists. PPD instantiates this policy via an offline-computed lookup table (Section 5); traditional PD is the special case $x \equiv 0$. Throughput is a system-level emergent metric and is not optimized per-request, but it improves as KV transfer volume drops (Table 2).

# 4. Interference Analysis and Comprehensive Trade-Offs

In this section, we begin by establishing a key empirical result that underpins the benefit of routing AP to D: Append-prefill causes an order-of-magnitude smaller interference than full prefill, making it feasible for decode nodes to handle Turn 2+ locally. We then conduct a comprehensive benchmark spanning 3,060 configurations to evaluate the effectiveness of Full AP-to-D configurations ($x{=}1$) and characterize their trade-offs. Throughout this section, we analyze static deployments where decode nodes with $x{=}1$ always handle Turn 2+ locally; Section 5 extends this to PPD, our dynamic per-request routing system.

## 4.1. Quantifying Prefill Interference

The core assumption behind PD disaggregation is that *all* prefill operations interfere severely with decode, necessitating physical isolation. Recent work (Gao et al., 2025; Shi et al., 2025) has begun to challenge this assumption by exploring adaptive isolation strategies. We further this direction by distinguishing two types of prefill.

**Full Prefill vs. Append Prefill.** *Full prefill* processes a new prompt without any cached context, the standard Turn 1 scenario with $O(n^2)$ attention complexity for $n$ input tokens. *Append prefill* processes only new tokens while

reusing cached KV from previous turns. For Turn 2+ with $m$ new tokens appended to the context of $n$ cached tokens, append-prefill computes attention only for the $m$ new tokens (each attending over $n + m$ keys), yielding $O(m(n + m))$ complexity. When $m \ll n$ (typical for follow-ups), append-prefill is roughly $n/m$ times cheaper than full prefill of the same total sequence length.

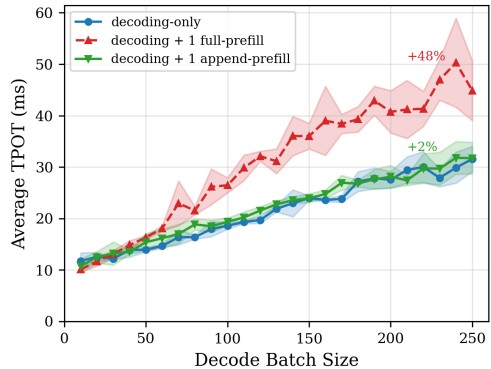

*Figure 2.* **Prefill-decode interference.** Decode TPOT degradation when co-locating with one full-prefill vs. one append-prefill operation (both processing 1,024 tokens). Full prefill causes a significant slowdown; append-prefill remains close to baseline. See Figure 7 in the Appendix for 4-prefill experiments showing consistent trends.

**Interference Measurement.** We conduct controlled micro-benchmarks on a single H100 GPU using Llama-3.1-8B to isolate interference effects, following established roofline analysis methodology (Yuan et al., 2024). We measure decode TPOT degradation when co-locating with full or append-prefill (Figure 2). Full prefill causes ∼48% slowdown at batch size 200; append-prefill causes only ∼2%, an order of magnitude less. With 4 concurrent prefills (see Figure 7), full reaches +57% while append stays at +21%, so the gap persists under increased concurrency.

This gap persists across context lengths: full prefill interference grows to 3–4× at 32K tokens while append-prefill stays below 25% even at 64K (Figure 8 in Appendix). These results confirm that decode nodes can safely handle Turn 2+ locally.

## 4.2. Configuration Space and Methodology

In this subsection, we describe the configuration space and methodology for systematically evaluating how different P/D assignments and routing parameters affect multi-turn serving performance.

**Machine Types and Routing Parameter.** We define three machine types: **P** (prefill-only), **D** (decode), and **R** (replica). The routing parameter $x \in [0, 1]$ determines D's Turn 2+ behavior: when $x{=}0$, D receives KV transfer from P every

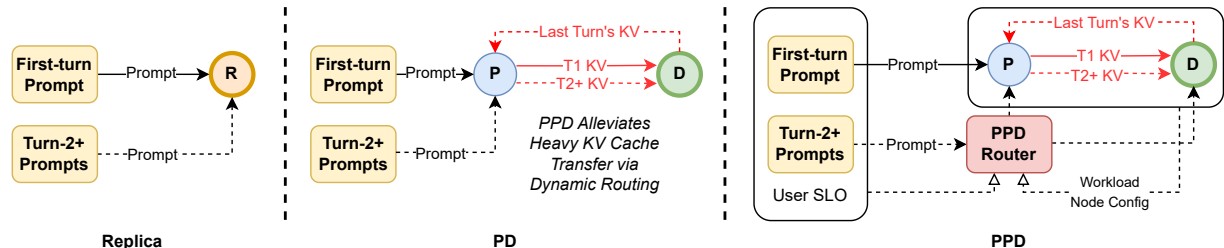

*Figure 3.* **Dynamic routing of append-prefill with PPD.** We illustrate the core concept behind PPD: **PPD** (Right) dynamically routes the append-prefill operations for multi-turn conversations to the prefill or decode nodes based on user SLO, estimated workload from the system, and the initial node configuration. Compared to **Replica** (Left), PPD retains all the benefits of disaggregation. In contrast to **PD** (Middle), PPD alleviates heavy KV cache transfer as well as extra recomputation for append-prefill with local cache, and can always adjust to meet various serving requirements. We want to highlight that both PD and any strategies with a fixed $x\%$ of append-prefill routed to decode nodes are special cases of our PPD.

turn (traditional PD); when $x=1$, D processes new input tokens locally using cached prefix (Full AP-to-D), eliminating network transfer overhead. Figure 3a illustrates how these routing schemes operate: Replica (local execution), PD ($x=0$: P→D with KV transfer every turn), and Full AP-to-D ($x=1$: P→D with KV transfer only on Turn 1).

**Configuration Space.** Using these three machine types with 4 GPUs, we explore 17 configurations, including 7 hybrid configurations that combine R with P/D (see Table 4 in Appendix). However, since hybrid configurations generally underperform pure PD alternatives and are orthogonal to AP routing optimization, we concentrate on 10 core configurations: **Replica** (1 config: all local execution), $\boldsymbol{x=0}$ configurations (3 configs: traditional PD with KV transfer every turn), $\boldsymbol{x=1}$ configurations (3 configs: Full AP-to-D with local Turn 2+ processing), and $\boldsymbol{0 < x < 1}$ (3 configs: partial AP routing to D, e.g., $x=\frac{1}{3}, \frac{1}{2}, \frac{2}{3}$).

**Workloads.** We design 18 synthetic workloads by combining 2 Turn 1 input/output length settings with 9 Turn 2 settings. Each setting specifies a fixed $(n_{\text{in}}, n_{\text{out}})$ pair, the number of input and output tokens for that turn. The 9 Turn 2 settings span three categories: *decode-heavy* (4 settings with short input and long output, e.g., $n_{\text{in}}=32, n_{\text{out}}=512$), *balanced* (2 settings with similar lengths), and *prefill-heavy* (3 settings with long input and short output, e.g., $n_{\text{in}}=1024, n_{\text{out}}=32$), covering the full spectrum of multi-turn conversation patterns.

**Experimental Setup.** All experiments were run on an internal cluster node with 4 NVIDIA H100 80GB HBM3 GPUs connected via NVLink. We evaluate the impact of slower inter-node interconnects via bandwidth simulation in Section 6.4. We use Llama-3.1-8B as the primary model, with validation experiments on Qwen2.5-14B and Qwen3-30B-A3B. Each configuration is evaluated at 10 QPS (queries per second) levels (0.5, 1, 2, 4, 6, 8, 10, 12, 16, 20), yielding $17 \times 18 \times 10 = 3{,}060$ data points. For each

test point, we run a fixed 10-second duration with conversations arriving according to a Poisson process at the target QPS (e.g., QPS=4 generates ∼40 two-turn conversations). We measure Turn 1 TTFT, Turn 2 TTFT, average TPOT, throughput (tokens/sec), and success rate.

### 4.3. Full AP-to-D Advantage: Eliminating KV Transfer Overhead

We now quantify the advantage of $x=1$ configurations (Full AP-to-D) over traditional PD disaggregation ($x=0$). The core finding: **switching from $x=0$ to $x=1$ reduces Turn 2 TTFT by 48–73%** by eliminating KV transfer for follow-up turns. Table 1 compares matched configurations: $x=1$ consistently outperforms $x=0$ on Turn 2 TTFT, with 1P_3D achieving up to 73.3% improvement and even 3P_1D showing 24.9–44.3% gains.

*Table 1.* Turn 2 TTFT improvement when switching from $x=0$ to $x=1$. Negative values indicate improvement. The $x=1$ advantage increases with load for P-scarce configurations (1P, 2P) but diminishes when P is abundant (3P).

| Config | Low QPS (0.5–2) | Med QPS (4–8) | High QPS (12–20) |
|---|---|---|---|
| 1P_3D: $x=0 \rightarrow x=1$ | −57.8% | −65.2% | **−73.3%** |
| 2P_2D: $x=0 \rightarrow x=1$ | −47.7% | −51.6% | −56.2% |
| 3P_1D: $x=0 \rightarrow x=1$ | −44.3% | −38.1% | −24.9% |

A striking pattern emerges: *the $x=1$ advantage increases with load*, since as QPS rises, transfer queuing grows and D's local cache access becomes increasingly valuable. The table also reveals that the $x=1$ advantage is maximized when P resources are scarce: 1P shows up to 73.3% improvement while 3P shows diminishing returns. In other words, **the fewer prefill nodes available, the greater the benefit of local processing on D**, because P becomes a bottleneck that $x=1$ bypasses entirely.

When P nodes are scarce, they become a bottleneck. In $x=0$ mode, Turn 2+ must traverse this bottleneck (P→D), while

*Table 2.* Winner distribution across three optimization objectives: minimizing Turn 2 TTFT, minimizing TPOT, and maximizing throughput. Each cell shows the percentage of (workload, QPS) combinations where that configuration category achieves the best result. Columns do not sum to 100% as hybrid configurations (which rarely win) are excluded. Full AP-to-D ($x=1$) configurations demonstrate the most balanced competitiveness, achieving the highest average win rate across all objectives.

| Mode | TTFT | TPOT | Thpt | Avg |
|---|---|---|---|---|
| Replica (4R) | **63.3%** | 0.6% | 0% | 21.3% |
| $x=0$ (PD) | 0% | **38.3%** | 4.4% | 14.2% |
| $0<x<1$ | 3.3% | 33.3% | 27.8% | 21.5% |
| $x=1$ (Full AP-to-D) | 27.2% | 15.6% | **38.3%** | **27.0%** |

$x=1$ bypasses P entirely. This makes $x=1$ configurations more resilient: at high QPS, they exhibit lower failure rates (see Table 5 in Appendix) since only Turn 1 contends for P capacity. When P is abundant, prefill capacity is no longer the bottleneck, and local AP overhead becomes relatively more significant.

**Validation: Turn and Model Scaling.** The $x=1$ advantage generalizes beyond 2-turn conversations and the primary 8B model. Scaling experiments (see Figure 10 in Appendix) show stable ~70% Turn 2+ TTFT improvement across 2–16 turns and three model sizes (8B, 14B, 30B), confirming that the benefit stems from architectural properties rather than model-specific characteristics.

### 4.4. No Universal Best: Objective-Dependent Optimization

The previous section established $x=1$'s clear advantage for Turn 2 TTFT. However, production LLM serving must balance multiple objectives: latency (TTFT), generation smoothness (TPOT), and system efficiency (throughput). When we expand our analysis to these dimensions, a more nuanced picture emerges.

**Core Finding.** *92.2% of workload-QPS combinations have different optimal configurations for Turn 2 TTFT versus Avg TPOT.* This fundamental tension means no single configuration dominates across all objectives. The optimal choice depends on the metric being optimized.

Table 2 quantifies this trade-off by showing the winner percentage for each configuration category across three objectives. Three patterns stand out:

*(1) Replica dominates Turn 2 TTFT.* With zero network transfer and local prefix caching, Replica achieves the lowest Turn 2 latency. However, Replica wins almost no TPOT or throughput scenarios: its lack of workload isolation lets prefill operations interfere with decode batches.

**Algorithm 1** PPD Dynamic Routing

1: **Phase 1: Offline Table Construction**
2: **for** each discretized workload $\hat{\psi}$ in benchmark grid **do**
3:   Measure Turn 2 TTFT and TPOT at $x=0$ and $x=1$
4:   Compute $S(\hat{\psi}; \pi, \mathbf{w})$ via Equation (1)
5:   Store $x^*(\hat{\psi}) \leftarrow \mathbb{1}[S > 0]$
6: **end for**
7:
8: **Phase 2: Online Per-Request Decision**
9: **Input:** request workload $\psi$ at turn $t$
10: **if** $t = 1$ **then return** $x = 0$ {no cached KV yet}
11: **end if**
12: Discretize $\psi$ to nearest grid entry $\hat{\psi}$
13: **return** $x^*(\hat{\psi})$

*(2) $x=0$ and $0<x<1$ configurations dominate TPOT.* Physical separation of prefill and decode workloads ensures stable token generation without interference. The decode-only (D) machines maintain consistent batch sizes and dedicated memory bandwidth, yielding predictable per-token latency.

*(3) Full AP-to-D ($x=1$) is the best disaggregated option for Turn 2 TTFT.* Among disaggregated configurations, $x=1$ captures over a quarter of Turn 2 TTFT wins while $x=0$ captures none. $x=1$ **also leads to throughput dominance** (38.3%): its cache reuse and reduced network load translate to higher system efficiency.

**Full AP-to-D's Balanced Profile.** A key observation from Table 2: $x=1$ (Full AP-to-D) is the *only* disaggregated mode that is competitive across all three objectives. While $x=0$ and $0<x<1$ configurations specialize in TPOT optimization, they rarely win on Turn 2 TTFT. $x=1$ achieves the highest average win rate across objectives (see Avg column), making it a robust default choice when workload characteristics are uncertain. Production deployments prioritizing smooth token delivery may still prefer $x=0$ configurations, which PPD recovers as the extreme weight setting (Section 6.5).

## 5. PPD: Dynamic AP Routing System

The trade-off analysis in Section 4.4 reveals that no single static $x$ dominates all objectives. We therefore design PPD, a dynamic routing system that selects $x$ per request from workload characteristics and operator weights, following workload-aware scheduling principles (Fu et al., 2024; Jain et al., 2025).

PPD operates in two phases (Algorithm 1). Offline, it builds a lookup table over a coarse workload grid by directly measuring TTFT and TPOT at the two extremes $x=0$ and $x=1$ for each cell, then storing the sign-of-score decision. Online, each Turn 2+ request is mapped to the nearest cell along

three axes (accumulated context length, input/output ratio, system QPS) and the precomputed decision is returned in $<1$ ms; Turn 1 always returns $x=0$ since no cached context exists. Discretization thresholds and the request feature schema appear in Section B.1.

**Decoupling Configuration Sizing from Turn 2+ SLO Tuning.** PPD splits the multi-turn serving control problem into two independent knobs. In traditional PD, the P:D ratio simultaneously dictates Turn 1 prefill capacity and Turn 2+ latency trade-offs, forcing the operator to balance a coupled multi-objective parameter. PPD decouples these: the P:D ratio governs Turn 1 throughput, while the operator weights **w** select the Turn 2+ operating point on the Pareto frontier. Empirically, PPD remains stable across all three P:D ratios we evaluate (Figure 4), whereas the static $x=0$ baseline collapses when P is scarce.

Our prototype builds on vLLM's disaggregated serving infrastructure, reusing its KV transfer protocol and prefix cache. Implementation details, including session management and routing protocol, are provided in Section B.

# 6. Real-World Validation

Synthetic workloads provide well-controlled conditions for systematically exploring the configuration space and isolating individual factors. To validate that these findings generalize to the heterogeneous distributions found in real traffic, we further evaluate PPD on real-world multi-turn conversation datasets.

## 6.1. Datasets and Setup

We use two publicly available multi-turn conversation datasets: **ShareGPT** (Chiang et al., 2023) with user-shared ChatGPT conversations with diverse topics and interaction styles, and **WildChat** (Zhao et al., 2024) with in-the-wild user conversations with varied prefill-to-decode ratios.

We filter multi-turn conversations from both datasets and evaluate three representative PD configurations (1P_3D, 2P_2D, 3P_1D) with and without dynamic PPD enabled at QPS levels from 1 to 20. The dynamic PPD mode uses balanced weights ($w_{\text{ttft}} = w_{\text{tpot}} = 1.0$). We measure Turn 2+ TTFT, average query latency, throughput, and success rate (defined as $\geq 95\%$ of requests completing without timeout).

## 6.2. PPD Improves Both Stability and Latency

Our real-world experiments (Figure 4) reveal two key findings: (1) PPD achieves consistently lower average query latency than baseline ($x=0$) across both datasets, and (2) PPD resolves the KV transfer bottleneck that causes $x=0$ mode degradation under multi-turn workloads. A complete three-way comparison including the $x=1$ static baseline is

provided in Section C.5.

**Finding 1: Lower Latency.** As shown in Figure 4, PPD curves (solid lines) consistently lie below baseline curves (dashed lines) across all configurations and QPS levels where both modes succeed. For the stable 1P_3D configuration on ShareGPT, PPD reduces average query latency by 15–25% across the QPS range. This improvement stems from PPD's elimination of redundant KV transfers for Turn 2+ requests: instead of traversing the P$\rightarrow$D network path every turn, follow-up requests execute locally on decode nodes using cached prefixes.

**Finding 2: Resolving Service Instability.** Beyond latency improvement, PPD transforms previously unusable configurations into stable deployments. The baseline ($x=0$) configurations exhibit service degradation (defined as success rate $<95\%$, primarily caused by request queuing and timeout under KV transfer saturation, not memory exhaustion or hardware failure): 2P_2D and 3P_1D's $x=0$ baselines degrade across the majority of QPS levels in both datasets, while only 1P_3D remains stable where sufficient D capacity absorbs the KV load.

In contrast, *PPD-enabled configurations achieve 100% success rate across all QPS levels and both datasets*. The $\times$ markers in Figure 4 visually demonstrate this gap: baseline curves are fragmented with numerous failure points, while PPD curves remain complete and continuous.

**Root Cause of Baseline Degradation.** The root cause is *KV transfer saturation*: in $x=0$ mode (traditional PD), decode nodes receive KV transfers for *every* turn, while PPD dynamically routes Turn 2+ requests with higher $x$ values, cutting KV transfer load by $\sim75\%$ at the observed average of 3.1 turns per conversation. At 3.1 average turns per conversation, this creates a $\sim3\times$ difference in network load, explaining why PD configurations degrade while PPD remains stable.

## 6.3. Comparison with the Strongest Static Baseline (x=1)

We next isolate PPD against the strongest static disaggregated baseline, $x=1$ (Full AP-to-D), the simplest heuristic that keeps every Turn 2+ request local. On end-to-end average latency the two appear visually similar (Figure 11 in Appendix), which is expected under balanced weights $w_{\text{ttft}}=w_{\text{tpot}}=1$: PPD trades some TTFT for TPOT improvement on requests it routes back to P, and these gains and losses cancel in the composite metric. A per-metric decomposition (Table 3) tells a different story.

PPD is the only mode that is competitive across all three axes: it beats $x=0$ on TPOT (12 vs. 10 wins) and beats $x=1$ on TTFT (14 vs. 13 wins), while matching $x=1$'s 100% suc-

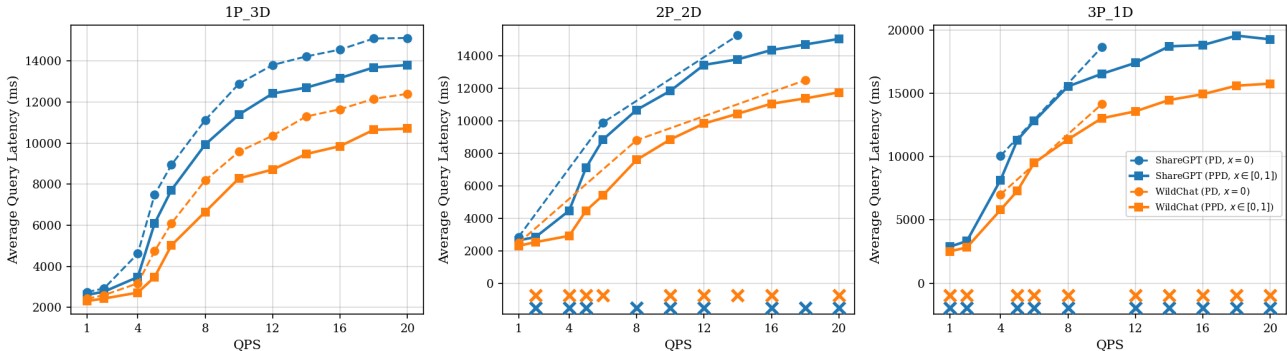

*Figure 4.* **PPD improves stability and reduces latency.** Average query latency vs. QPS for three configurations (1P_3D, 2P_2D, 3P_1D) on ShareGPT (blue) and WildChat (orange) datasets. Dashed lines: PD ($x$=0); solid lines: PPD ($x \in [0, 1]$). × **markers indicate service degradation** (success rate <95% due to request timeout). PPD consistently achieves lower latency than PD while maintaining stability across the entire QPS range.

*Table 3.* Per-metric winner counts across 27 test points (3 configs × 9 QPS, WildChat). PPD achieves the most TPOT wins (surpassing $x$=0) and the most TTFT wins (surpassing $x$=1), while matching $x$=1's 100% success rate. Neither static baseline dominates all metrics; PPD is the only mode competitive across all three.

| Mode | TPOT Best | TTFT Best | SR = 100% |
|------|-----------|-----------|-----------|
| $x$=0 | 10/27 | 0/27 | 4/27 |
| $x$=1 | 5/27 | 13/27 | 27/27 |
| **PPD** | **12/27** | **14/27** | **27/27** |

cess rate. Since $x$=1 is already the simplest static rule that exploits local KV at all, any weaker heuristic (e.g., random routing) is strictly dominated; the meaningful comparison is therefore whether PPD improves over $x$=1, which Table 3 confirms it does on each metric individually. The weights ($w_\text{ttft}$, $w_\text{tpot}$) are precisely the control knob: PPD recovers $x$=0 or $x$=1 as special cases at extreme settings, and Section 6.5 characterizes the smooth interpolation.

### 6.4. Robustness Across Network Speeds

Our experiments use intra-node NVLink, but production multi-node deployments rely on RDMA over InfiniBand or Converged Ethernet (RoCE), typically 2–8× slower (Patel et al., 2025; Qin et al., 2025). Since PPD bypasses the P→D channel for Turn 2+ requests, its advantage should grow as the network slows. To verify this without multi-node hardware, we follow the bandwidth-emulation methodology of TetriInfer (Hu et al., 2024b) and inject a calibrated extra delay on each PD-routed request, leaving PPD's local path untouched. The exact delay model and the per-token KV footprint are in Section B.6.

Figure 5 sweeps four interconnects from NVLink down to 100GbE. As effective bandwidth drops by an order of magnitude, the PD baseline's Turn 2+ TTFT rises by ∼19% while PPD stays at ∼51 ms; the relative TTFT reduction widens from 64% to 70%, with TPOT and end-to-end latency tracking the same trend. NVLink-based evaluation is therefore a conservative lower bound for PPD's advantage in real multi-node deployments.

### 6.5. Case Study: Weight-Based TTFT-TPOT Trade-Off

So far we ran PPD with balanced weights ($w_\text{ttft}$=$w_\text{tpot}$=1). To illustrate the operator-facing trade-off, we sweep $w_\text{tpot}$ on a slice of the workload where PPD is most challenged: 500 prefill-heavy multi-turn conversations sampled from ShareGPT and WildChat, run on 1P_3D at QPS 8 and 16. As $w_\text{tpot}$ rises, the decision engine increasingly prefers the PD path over local processing.

Figure 6 reveals that the weight ratio $w_\text{tpot}/w_\text{ttft}$ is a *monotonic control surface*: it smoothly interpolates between the $x$=1 regime (94–96% TTFT reduction, 7–12% TPOT degradation) and the $x$=0 baseline. Operators thus pick a single point on this surface offline, rather than tuning multiple coupled parameters online. PPD is a *routing actuator*: it does not enforce end-to-end SLO bounds (which require closed-loop admission control and batch scheduling), but exposes a predictable knob that higher-level SLO controllers can drive from observed P99 metrics.

## 7. Discussion

**Relationship with Concurrent Work and Limitations.** Concurrently, AMPD (He et al., 2026) identifies the same multi-turn inefficiency in PD disaggregation and proposes coordinating prefill workloads via real-time queue-state estimation paired with offline hardware planning. Our work is complementary in scope: PPD contributes (i) a micro-architectural account of *why* local append-prefill is viable, namely the order-of-magnitude interference gap (Section 4.1); (ii) a clean optimization formulation in which traditional PD is the special case $x \equiv 0$; and (iii) a single-knob

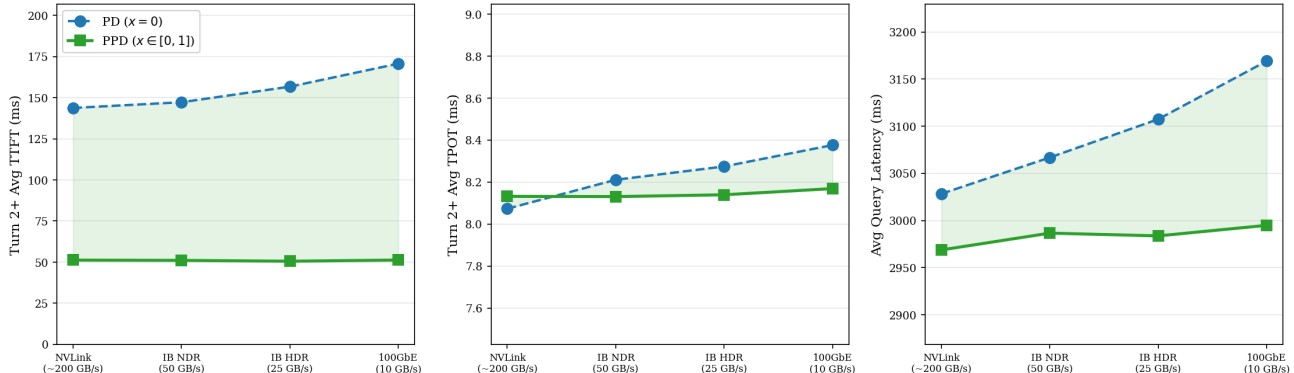

*Figure 5.* **PPD's advantage grows monotonically as the simulated network slows.** Results on 1P_3D, QPS=1, WildChat (500 conversations) across four interconnects: NVLink (∼150 GB/s effective, intra-node), InfiniBand NDR (50 GB/s), InfiniBand HDR (25 GB/s), and 100GbE (10 GB/s). **Left:** Turn 2+ TTFT. PD ($x$=0) rises from 143.7 ms to 170.6 ms (+18.7%) as bandwidth drops; PPD remains flat at ∼51 ms. **Center:** Turn 2+ TPOT. PD drifts from 8.07 ms to 8.38 ms (+3.8%) under network contention; PPD stays near 8.1 ms. **Right:** End-to-end latency. PD increases by 4.7% (3,028 ms → 3,169 ms); PPD by 0.9%, attributable solely to Turn 1.

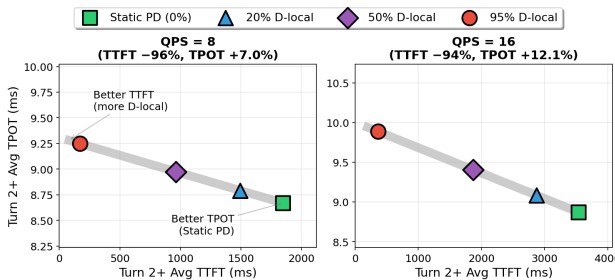

*Figure 6.* $w_{tpot}$ **traces a monotonic frontier between TTFT and TPOT.** Turn 2+ latency on 1P_3D under prefill-heavy workloads. Operating points (left to right): static PD (0% D-local); $w_{tpot}$=6 (20%); $w_{tpot}$=3 (50%); balanced $w_{tpot}$=1 (95%).

TTFT–TPOT control surface (Equation (1)) that operators can tune offline. The principal limitation of PPD is that its lookup table is built offline and degrades gracefully but suboptimally when hardware or workload distributions drift far from the calibration set; AMPD's online queue-state estimation is a natural mechanism for closing this gap, and we view PPD's principled foundation and AMPD's adaptive loop as composable rather than competing. Other natural extensions include integration with higher-level SLO controllers that drive **w** from observed P99 metrics, deployment on heterogeneous GPU clusters (Tong et al., 2025) or long-context elastic parallelism (Wu et al., 2024), and joint use with distributed KV cache layers (below).

**Relationship with Distributed KV Cache.** Distributed KV cache layers (e.g., Mooncake (Qin et al., 2025), MemServe (Hu et al., 2024a)) operate at a different layer than PPD: they decide *where* prefix states are stored across the cluster, while PPD decides *how* cache-hitting and cache-missing requests are scheduled to avoid mutual interference. The two are fully compatible. Distributed storage maxi-

mizes prefix reuse, while PPD ensures that warm requests are not delayed by cold ones; we expect joint deployment to amplify both effects in large-scale serving.

## 8. Conclusion

Multi-turn conversations expose fundamental limitations in PD disaggregation that single-turn workloads do not reveal. The order-of-magnitude interference gap between full prefill and append-prefill opens a new design dimension: decode nodes can safely handle Turn 2+ requests locally. Yet our systematic exploration of 3,060 configurations confirms that static routing cannot universally optimize TTFT, TPOT, and throughput. PPD realizes a workload-aware alternative through an offline-tuned, single-knob control surface (Equation (1)), cutting Turn 2+ TTFT by an average of ∼68% on real-world workloads while keeping TPOT competitive and adding <1 ms of per-request overhead.

## Impact Statement

This work improves the efficiency of multi-turn LLM serving, potentially reducing computational costs and energy consumption in large-scale deployments. We do not foresee direct negative societal impacts beyond those inherent to LLM technology broadly.

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

# A. Configuration Space

All configurations use a fixed budget of 4 GPUs. Notation: xP = x Prefill-only nodes, xD = x Decode nodes, xR = x Replica nodes. The routing parameter $x \in [0, 1]$ determines D's Turn 2+ behavior.

**Why Hybrid Configurations Are Excluded from Main Analysis.** Our experiments include 7 hybrid configurations that combine R (Replica) nodes with P/D nodes, as listed in Table 4. However, we exclude hybrid configurations from our main analysis for two reasons.

First, R nodes do not benefit from disaggregation's workload isolation. Unlike dedicated P and D nodes, an R node handles both prefill and decode operations on the same GPU. When R processes a prefill operation, it interferes with its own ongoing decode batches, the very problem that PD disaggregation was designed to solve. This means the routing parameter $x$ (which controls AP routing to D nodes) does not apply to R nodes, making hybrid configurations orthogonal to our focus on AP routing optimization.

Second, hybrid configurations reduce the pool of dedicated P/D resources. For example, 1R_1P_2D allocates only 1 GPU to prefill (vs. 2 in 2P_2D), creating potential prefill bottlenecks under high load. In our experiments, hybrid configurations rarely achieved the best performance on any metric, winning only 6.1% of TTFT, 12.2% of TPOT, and 16.1% of throughput test points.

# B. Implementation Details

This appendix provides implementation details for the PPD routing system described in Section 5.

### B.1. Decision-Engine Discretization

The PPD lookup table indexes Phase-1 measurements along three discrete axes:

- *Context class* bins the accumulated context length $n_{\text{ctx}}$ (cumulative tokens from prior turns) into small ($\leq 512$), large (512–4096), and huge ($>4096$) tokens.
- *Workload type* classifies the input/output token ratio $n_{\text{in}}/n_{\text{out}}$ into nine categories spanning decode-heavy, balanced, and prefill-heavy regimes (matching the synthetic grid in Section 4.2).
- *QPS bin* snaps the current system QPS to the nearest benchmark level used in Phase 1.

At inference time, an incoming request with feature tuple $\psi = (t, n_{\text{in}}, n_{\text{out}}, n_{\text{ctx}}, q)$ (turn number, new-input tokens, expected output tokens, accumulated context, current QPS) is mapped to the nearest grid cell $\hat{\psi}$ along these axes, and the precomputed $x^*(\hat{\psi})$ is returned.

### B.2. KV Transfer Protocol

Our implementation reuses vLLM's disaggregated serving infrastructure. P nodes run with `kv_role=kv_producer`, generating KV caches and sending them via ZeroMQ. Decode servers run with `kv_role=kv_consumer`, receiving KV caches and storing them in local prefix cache for Turn 2+ processing when $x>0$. The transfer protocol follows vLLM's standard disaggregated serving format, requiring no custom modifications.

### B.3. Session Management

The routing proxy maintains a session table as an in-memory dictionary:

```
session_table[conv_hash] = {
    "turn_count": int,
    "assigned_pd": str,
    "last_access": timestamp
}
```

Conversation identifiers are computed as MD5 hashes of the first user message. Sessions are evicted after 60 minutes of inactivity, aligning with vLLM's default prefix cache TTL.

### B.4. Service Discovery

Backend servers register with the proxy via ZeroMQ heartbeats every 10 seconds. The proxy maintains instance lists for each server type (P, D, R) and removes servers after 30 seconds without a heartbeat. This enables dynamic scaling and automatic failure recovery.

### B.5. Proxy Implementation

The routing proxy is implemented using Quart (async Flask) and handles:

- Request parsing: Extract conversation context and determine turn number

- Session lookup/creation: Manage conversation-to-server mappings

- Request forwarding: Stream responses from backend to client

- Statistics collection: Track routing decisions and latency metrics

### B.6. Bandwidth Simulation

To emulate slower interconnects without multi-node hardware (Section 6.4), we inject a calibrated extra delay on the decode-side receive path of vLLM's P2P NCCL connector.

*Table 4.* Complete list of 17 configurations evaluated in our experiments. The routing parameter $x$ determines the fraction of AP operations routed to D nodes.

| Category | Configurations | $x$ | Count |
|---|---|---|---|
| Replica | 4R | N/A | 1 |
| $x{=}0$ (PD) | 1P_3D, 2P_2D, 3P_1D | 0 | 3 |
| $x{=}1$ (Full AP-to-D) | 1P_3D, 2P_2D, 3P_1D | 1 | 3 |
| $0{<}x{<}1$ (partial AP routing) | 1P_3D, 2P_2D | $\frac{1}{3}, \frac{2}{3}, \frac{1}{2}$ | 3 |
| Hybrid | 1R_1P_2D, 1R_2P_1D, 2R_1P_1D, etc. | 0, 1, $\frac{1}{2}$ | 7 |

For each PD-routed request, the injected delay is

$$\Delta t = \max\left(0, \frac{B(\psi)}{\beta_{\text{target}}} - t_{\text{NVLink}}\right), \qquad (2)$$

where $B(\psi) = n_{\text{tokens}} \cdot s_{\text{kv}}$ is the request's KV cache footprint, $\beta_{\text{target}}$ is the simulated bandwidth, and $t_{\text{NVLink}}$ is the actual NVLink transfer time (subtracted to avoid double-counting). For Llama-3.1-8B (BF16, GQA with 8 KV heads, 32 layers) the per-token footprint is $s_{\text{kv}} = 128$ KiB; at the WildChat P90 context length of 5,115 tokens, each Turn 2+ transfer in the $x{=}0$ path moves ∼670 MB, about ∼4.5 ms over NVLink, ∼27 ms over InfiniBand HDR (25 GB/s), and ∼67 ms over 100GbE. PPD's local append-prefill path is unaffected by the injection.

## C. Additional Experimental Results

### C.1. Extended Interference Analysis

The main paper (Figure 2) demonstrates the interference gap between full prefill and append-prefill with a single concurrent prefill operation. Here we extend this analysis to examine robustness under higher concurrency and longer context lengths.

Figure 7 increases concurrency to 4 simultaneous prefill operations. The interference gap persists: full prefill causes ∼57% TPOT degradation at batch size 200, while append-prefill remains within ∼21% of baseline. This confirms that the fundamental difference in interference characteristics is not an artifact of low concurrency.

Figure 8 varies context length from 2K to 64K tokens. Full prefill interference grows quadratically, reaching 3–4× slowdown at 32K tokens. In contrast, append-prefill interference remains below 25% even at 64K tokens, validating that append-prefill scales linearly with context length.

### C.2. Pareto Analysis

Figure 9 presents the complete Pareto frontier analysis across all workload types and QPS levels evaluated in our benchmark. Each subplot shows P99 TTFT versus throughput (TPS) for the 10 core configurations; the visualization

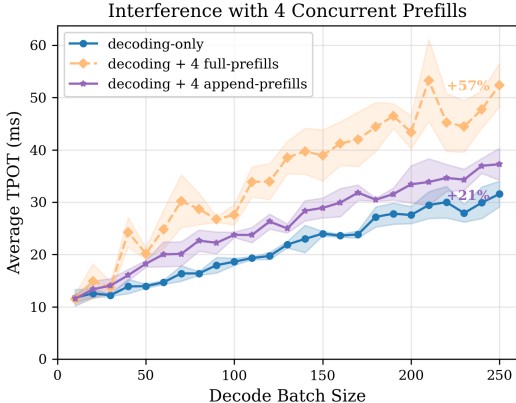

*Figure 7.* **Prefill-decode interference with 4 concurrent prefills.** Same setup as Figure 2 but with 4 concurrent prefill operations. Full prefill causes ∼57% slowdown at batch size 200; append-prefill remains within ∼21% of baseline.

complements the aggregate winner-distribution in Table 2 by exposing panel-level structure.

Two structural patterns deserve attention. Reading across columns (decode-heavy, balanced, prefill-heavy), the absolute TTFT scale grows because prefill-heavy mixes amplify both the KV transfer cost on the $x{=}0$ side and the append-prefill cost on the $x{=}1$ side, widening the spread of frontier points by roughly an order of magnitude between leftmost and rightmost columns at fixed QPS. Reading down rows (QPS=2, 8, 20), the orange baseline points fan out toward higher TTFT as transfer queuing accumulates, while the blue D-local-capable points stay clustered closer to the lower-TTFT region, particularly under decode-heavy mixes where the local prefix cache is hottest and append-prefill is cheapest.

The 9-panel grid refines the no-universal-best finding (Section 4.4): even within the two-axis TTFT vs. TPS view, no single configuration occupies every Pareto frontier. The optimum slides toward $x{=}1$ as QPS rises and the workload becomes more decode-heavy, while balanced and prefill-heavy panels at low QPS leave room for $x{=}0$ to compete on TPS. PPD's per-request choice rule (Equation (1)) is the

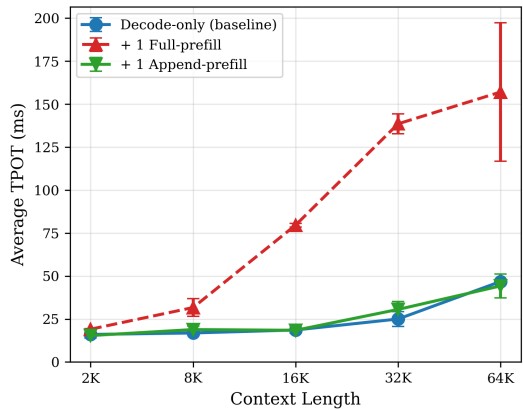

*Figure 8.* **Interference scaling with context length.** Full prefill interference grows to 3–4× at 32K tokens; append-prefill stays below 25% even at 64K.

natural way to reconcile this: it picks the locally favorable point on each panel rather than committing to one globally.

### C.3. Scaling Validation

A natural question is whether the $x{=}1$ advantage observed in our 8B experiments generalizes to longer conversations and larger models. Figure 10 addresses this by measuring Turn 2+ TTFT across (a) 2–16 turn conversations and (b) 8B–30B model sizes.

The results confirm that $x{=}1$ maintains its advantage across both dimensions. For turn scaling, the TTFT gap between $x{=}0$ and $x{=}1$ widens as conversations grow longer, because accumulated context amplifies KV transfer overhead. For model scaling, the relative improvement remains stable at ∼70%, indicating that the benefit stems from architectural properties rather than model-specific characteristics.

Our comprehensive experiments focus on 8B models for practical reasons: replica configurations require the full model to fit on a single GPU, and each configuration switch requires a complete server restart. Testing 17 configurations across 180 workload-QPS combinations with larger models would incur prohibitive initialization overhead.

### C.4. Failure Rate Analysis

Beyond latency metrics, system reliability is critical for production deployments. Table 5 reports request failure rates (timeout $> 30s$) across configurations at high QPS levels.

Two patterns emerge. First, configurations with extreme P:D ratios (3P_1D) are inherently fragile: with only one decode GPU, any queuing at the decode stage cascades into timeouts. Second, within each P:D ratio, $x{=}1$ consistently achieves lower failure rates than $x{=}0$. For example, 2P_2D

with $x{=}1$ maintains 0% failure rate at QPS=12 where $x{=}0$ already shows 6% failures. This reliability advantage stems from reduced network contention: $x{=}1$ eliminates Turn 2+ KV transfers, freeing bandwidth for Turn 1 transfers that cannot be avoided.

*Table 5.* Failure rates at high QPS. $x{=}1$ configurations consistently show lower failure rates than their $x{=}0$ counterparts.

| Config | QPS=8 | QPS=12 | QPS=16 | QPS=20 |
|---|---|---|---|---|
| 3P_1D ($x{=}0$) | 11% | 44% | 61% | 89% |
| 3P_1D ($x{=}1$) | 6% | 22% | 44% | 67% |
| 2P_2D ($x{=}0$) | 0% | 6% | 11% | 22% |
| 2P_2D ($x{=}1$) | 0% | **0%** | 6% | 11% |
| 4R | 0% | 0% | 0% | 0% |

### C.5. Three-Way Comparison: $x{=}0$ vs. $x{=}1$ vs. PPD

Figure 11 presents the complete three-way comparison on WildChat across all configurations and QPS levels; the per-metric decomposition is reported as Table 3 in the main text (Section 6.3).

Reading the panels left to right (1P_3D, 2P_2D, 3P_1D), several patterns are visible. The static $x{=}0$ baseline (blue dashed) accumulates failure markers at progressively lower QPS as the P:D ratio shifts away from the well-balanced 1P_3D, reflecting the diminishing decode capacity to absorb continuous KV transfers; in 3P_1D, the single decode GPU becomes the bottleneck and a substantial fraction of the QPS range falls below the SR≥95% threshold. By contrast, both $x{=}1$ and PPD achieve a 100% success rate across all 27 test points and all three configurations, and their E2E latency curves are visually almost indistinguishable across the entire QPS sweep.

This visual similarity is precisely what motivates the per-metric decomposition in the main text. Under balanced weights $\mathbf{w} = (1, 1)$, PPD trades a small amount of TTFT for TPOT improvement on the subset of requests it routes back through P, and these two effects nearly cancel in the composite E2E metric, producing curves that overlay $x{=}1$'s. The per-metric numbers in Table 3 disentangle this: PPD beats $x{=}0$ on TPOT (12 vs. 10 wins out of 27) and beats $x{=}1$ on TTFT (14 vs. 13 wins), while matching $x{=}1$'s perfect success rate. PPD does not pay a stability cost for its dynamism (Figure 11); the per-metric numbers in Table 3 confirm that dynamism buys improvements no single static $x$ delivers simultaneously.

A practical reading of the three panels is that the relative value of dynamic routing scales with how stressed the decode pool is. In 1P_3D, the abundant decode capacity makes $x{=}1$ already sufficient and PPD's per-request decisions only marginally adjust the trajectory. In 2P_2D, the two curves remain close because the decode load is moderate. In 3P_1D,

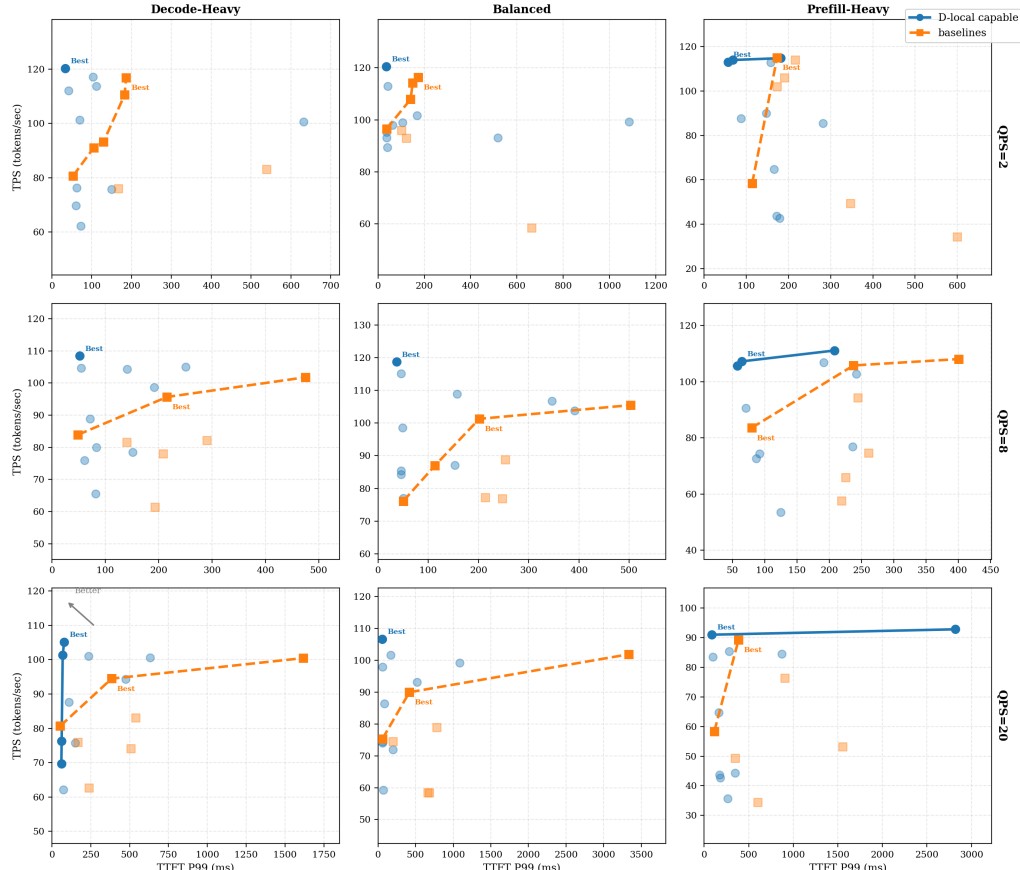

*Figure 9.* **P99 TTFT vs. TPS Pareto frontiers.** Columns: Decode-heavy, Balanced, Prefill-heavy workloads. Rows: QPS=2, 8, 20. D-local capable configurations (blue) achieve lower TTFT while maintaining competitive TPS, with the advantage most pronounced under high QPS and decode-heavy workloads.

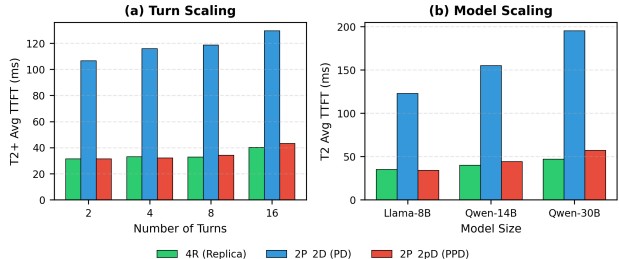

*Figure 10.* $x{=}1$ **maintains advantage across turns and model sizes.** (a) Turn scaling: Turn 2+ TTFT across 2–16 turns. (b) Model scaling: Turn 2 TTFT across 8B–30B models. $x{=}1$ closely matches Replica performance while preserving disaggregation benefits; $x{=}0$ degrades sharply with context growth.

where the lone decode node becomes the contention point under high QPS, PPD has the most room to redistribute Turn 2+ load back to the abundant P pool, narrowing the gap between TTFT-side and TPOT-side metrics. This pattern is consistent with the score-based decision rule in Equation (1): route locally when local processing is cheap, and fall back to P when it is not.

The 30 s timeout threshold underlying the failure markers is a deliberately conservative SLO. Tightening it to 10 s shrinks the SR= 100% region for $x{=}0$ further but leaves $x{=}1$ and PPD unchanged across the entire QPS range, because the local-cache path keeps Turn 2+ end-to-end latency well below either threshold even at QPS=14. We chose the looser threshold to make the failure mode visible across the full sweep rather than collapsing every high-QPS $x{=}0$ point into an opaque "failed" marker; readers prioritizing tighter SLOs can read Figure 11 as an upper bound on $x{=}0$'s viable operating envelope.

It is also worth relating Figure 11 to the bandwidth-simulation results in Section 6.4. Figure 5 fixes QPS at 1 and varies the simulated network bandwidth, isolating the per-transfer cost component of PPD's advantage; Figure 11 fixes bandwidth at intra-node NVLink and sweeps QPS, isolating the queuing-time component. Together the two experiments establish that PPD's gain over $x{=}0$ has two distinct origins: a static, per-request transfer-time saving that

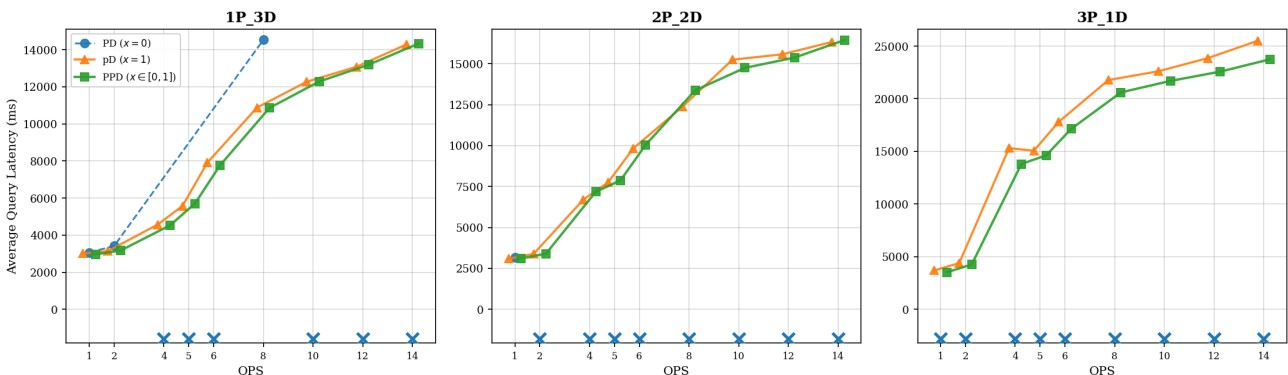

*Figure 11.* Three-way E2E latency comparison on WildChat (3 configs × 9 QPS). PD ($x$=0, blue dashed): fails catastrophically (×markers, SR<95%) in 13/27 test points. pD ($x$=1, orange triangle) and PPD (green square): both maintain stability with near-identical E2E curves. The E2E similarity between $x$=1 and PPD is expected under balanced weights ($w_{\text{ttft}}$=$w_{\text{tpot}}$=1.0): PPD trades some TTFT for TPOT improvement on routed-back requests, which cancels out in the composite E2E metric. The per-metric decomposition in Table 3 reveals that PPD breaks both static baselines' per-metric monopolies.

grows with slower interconnects, and a dynamic, queuing-time saving that grows with concurrent load. Neither is recoverable for $x$=0 once the deployment is in place, which is why even the multiplicative composition of small per-mechanism gains shows up so prominently in the aggregate E2E curves.

Finally, the 27 test points in this experiment are not chosen adversarially: each lies within the SLO envelope a reasonable operator might target, with QPS values bracketing typical conversational traffic on a 4-GPU node and conversation lengths drawn directly from real WildChat sessions. The systematic shift from $x$=0 failures (13/27) toward universal $x$=1 and PPD success (27/27) is therefore not a worst-case demonstration but a representative one. Combined with the per-metric decomposition in Table 3, it is the strongest single piece of evidence we present that dynamic routing is preferable to any static $x$ in production multi-turn workloads.

