# OpenReview forum: "Not All Prefills Are Equal: PPD Disaggregation for Multi-turn LLM Serving"
_ICML.cc/2026/Conference — ICML 2026 regular_

### Official Review · Reviewer_Erft · 2026-02-21

**Soundness:** 4
**Presentation:** 3
**Significance:** 3
**Originality:** 3
**Overall Recommendation:** 4
**Confidence:** 3

**Summary:**

This paper addresses the overhead of KV cache migration and frequent network transfers in multi-turn LLM serving under PD disaggregation.
The authors observe that the impact of append-prefill on the decoding process is relatively minor, suggesting that such prefill tasks can be handled directly on decoding instances. To leverage this, the paper proposes PPD, a dynamic append-prefill routing system that determines where to execute append-prefill based on SLOs. Experimental results demonstrate that PPD significantly reduces the TTFT for subsequent turns in a conversation while maintaining a low TPOT.

**Compliance With Llm Reviewing Policy:**

Affirmed.

**Final Justification:**

I have read the authors' rebuttal, which thoroughly addressed my main concerns.

I maintain my score of 4 (Weak Accept). The paper is technically sound with excellent experimental rigor, addressing a real and increasingly important problem in disaggregated LLM serving. The key insight — that append-prefill can be routed to decoding instances with minor interference — is well-validated, and the dynamic routing framework is a practical contribution. The framing of PPD as a smooth interpolation on the TTFT-TPOT Pareto frontier is elegant. My only lingering note is that the core assumption, while validated extensively, is inherently workload-dependent, and broader deployment experience would strengthen the claim.

**Key Questions For Authors:**

1. A fundamental assumption of PPD is that append-prefill does not significantly interfere with decoding in most cases. Regarding the experiments shown in Figure 2, what was the specific token count for the "append" part? In practical conversational scenarios, the number of newly appended tokens can often exceed the length of the existing context. Does the assumption still hold in such cases?
2. Given that multi-turn dialogues introduce performance penalties in PD disaggregated setups, I would like to see a comparison with a non-disaggregated baseline.
3. The experiments were conducted on single-GPU deployments using NVLink for KV cache transfer. However, in production-scale PD disaggregation, KV cache migration typically occurs over RDMA. How would the difference in bandwidth and latency between NVLink and RDMA affect your experimental conclusions?
4. In standard LLM deployments, the ratio of prefill to decode instances is typically tuned based on SLOs. Does the introduction of PPD’s dynamic routing make the optimal PD ratio significantly more difficult to determine or stabilize?

**Limitations:**

yes

**Strengths And Weaknesses:**

Strengths
1. The paper tackles a critical issue in PD disaggregation: the significant KV cache transfer and storage overhead inherent in multi-turn LLM inference.
2. The authors provide comprehensive experimental evaluations to validate the effectiveness of the proposed PPD system.

Weaknesses
1. The core assumption of PPD that append-prefill has minor impact on decoding may not hold universally across all workloads or model architectures.

---

> ### Author Rebuttal · Authors · 2026-03-30
>
> Thank you for your positive assessment and insightful questions — particularly Q4 on PD ratio tuning, which pushed us to articulate an aspect of PPD's design that we had not fully elaborated in the paper!! We address each question below.
>
> **Q1: Append-prefill token count and assumption validity.** The interference benchmark in Figure 2 uses 1024 tokens for both full and append prefill, as stated in the caption. And we explicitly validate that the interference gap holds at much larger scales: Figure 7 (Appendix) varies context length from 2K to 64K tokens, showing that full prefill interference grows to 3–4× at 32K while append prefill stays below 25% even at 64K. Figure 6 (Appendix) confirms the gap persists with 4 concurrent prefills (+57% for full vs +21% for append). Figure 9 validates across model sizes (8B, 14B, 30B) and 2–16 turns, with stable ~70% TTFT improvement.
>
> For cases where appended tokens exceed cached context (m > n), append prefill approaches full prefill cost by definition — but this is precisely what PPD's decision engine handles. Algorithm 1 classifies each request by its actual (n_in, n_out, n_ctx) and routes large-input requests back to P when TPOT penalty outweighs TTFT benefit. PPD does not assume small appends; it adapts to the actual workload.
>
> **Q2: Non-disaggregated baseline comparison.** Replica (4R) is our non-disaggregated baseline, already included in Figure 1 and Table 2. It is a strong baseline: Table 2 shows Replica dominates Turn 2 TTFT (63.3% win rate) thanks to zero network transfer and local prefix caching. However, Replica wins almost no TPOT or throughput scenarios (0.6% and 0%) because co-locating prefill and decode on the same GPU causes severe interference (Section 4.1).
>
> This is precisely the trade-off PPD navigates. Our core finding (Section 4.4) is that no single configuration — including Replica — dominates all SLOs simultaneously. PPD allows operators to balance this trade-off via weights (w_ttft, w_tpot): increasing w_ttft pushes the system toward more local processing (approaching Replica's TTFT advantage), while increasing w_tpot favors routing through P (preserving disaggregation's TPOT benefit). Figure 5 demonstrates this smooth interpolation on the TTFT-TPOT Pareto frontier. In this sense, PPD unifies disaggregated and non-disaggregated serving as endpoints of a continuous spectrum, rather than forcing a binary choice.
>
> **Q3: NVLink vs RDMA.** We address this shared concern comprehensively in our response to Reviewer ApRF (Q2), including theoretical analysis, bandwidth simulation experiments, and supplementary Figure [link](https://anonymous.4open.science/r/anonymous-4217/scaling.png). Please refer that for more information.
>
> In brief: PPD's advantage grows monotonically with slower interconnects because it eliminates Turn 2+ KV transfers entirely. Our NVLink evaluation is a conservative lower bound — simulation across NVLink/IB NDR/IB HDR/100GbE confirms TTFT reduction grows from 64.4% to 69.9% as bandwidth decreases, while PPD's latency remains invariant. Please see that response for full details and the supplementary figure.
>
> **Q4: Does PPD complicate PD ratio tuning?** We argue PPD simplifies it. In traditional PD, the PD ratio simultaneously affects both Turn 1 and Turn 2+ performance — a coupled optimization problem. Adding P nodes improves Turn 2+ TTFT (more prefill capacity) but reduces D nodes, degrading TPOT. Operators must jointly balance TTFT and TPOT across the full multi-turn lifecycle.
>
> PPD decouples this. Turn 1 still flows through P→D, so the PD ratio governs Turn 1 latency as before. But Turn 2+ routing is managed independently by (w_ttft, w_tpot), which controls the TTFT-vs-TPOT trade-off for subsequent turns. This gives operators two independent knobs — PD ratio for Turn 1 capacity, weights for Turn 2+ SLO balance — rather than one coupled multi-objective parameter.
>
> Our experiments support this: PPD achieves stable Turn 2+ performance across all three PD ratios (1P:3D, 2P:2D, 3P:1D) and maintains 100% success rate regardless of ratio (Figure 4, Section 6.2), whereas x=0 degrades catastrophically when P is scarce. The PD ratio becomes primarily a Turn 1 sizing decision, making the system more predictable and easier to tune.

---

> > ### Author Rebuttal · Reviewer_Erft · 2026-04-01
> >
> > Thank you for the thorough rebuttal. My main concerns — particularly regarding the generality of the append-prefill assumption (Q1) and the comparison with non-disaggregated baselines (Q2) — have been adequately addressed. I will maintain my positive score.

---

> > > ### Author Response · Authors · 2026-04-02
> > >
> > > Thank you for taking the time to read our rebuttal carefully and for confirming that your concerns have been fully resolved! We hope the new experiments and clarifications added during rebuttal give you further confidence in the paper's contributions. We truly appreciate your constructive and thoughtful engagement throughout the review process.

---

### Official Review · Reviewer_1txr · 2026-03-04

**Soundness:** 3
**Presentation:** 2
**Significance:** 3
**Originality:** 3
**Overall Recommendation:** 4
**Confidence:** 3

**Summary:**

This paper proposes PPD, an extension of PD disaggregation in large-scale LLM inference systems that dynamically routes prefill requests that reuse prior rounds' KVs to decode instances (as opposed to prefill instances) to eliminate the network traffic and efficiently reuse the KVs. PPD reduces turn-2 TTFT by ~70% while maintaining similar TPOT.

**Compliance With Llm Reviewing Policy:**

Affirmed.

**Final Justification:**

I appreciate the reviewers for clarifying my questions. I maintain my stance that the solution, as the authors transparently acknowledged, has limitations. However, I do believe that the paper sheds light on interesting insights that might motivate future research. As a result, I have increased my recommendation from a weak reject to a weak accept.

**Key Questions For Authors:**

### 1. Could the authors justify the design choice of the baseline PD disaggregation scheme?

The authors mention in the abstract that: "every turn requires prefilling the new prompt and response from the last turn". Why does the response need prefilling if their KV cache is already generated by the decode node? Could the authors clarify whether "regeneration" is preferred over "KV cache transmission and reusing" for any particular reason? Are there relevant references that suggest this is the default behavior of PD disaggregation in the wild?

In 4.2, if x=0, does P take any KV transfers from D? Does P, the prefill node, always do regeneration (prefilling the whole sequence), or is it partial generation? If it's the latter, does the prefill node only prefill the new input, or the output's KVs from the last round as well?

### 2. The authors claim that PPD adapts to varying SLOs. However, is it true that this process is not explicit/direct?

On a high level (especially in the systems diagram in Fig. 3), PPD takes into consideration the user's SLO specifications. However, in the routing mechanism, there's no interface for users to specify their SLOs on TTFT/TPOT. It seems that operators need to manually fine-tune the w_ttft and w_tpot weights to adjust the score, which is a very indirect and manual process. Typically, systems take in operators' SLO constraints (e.g., >X % of the queries need to have TTFT <Y ms). Even though PPD exposes knobs that allow operators to influence the system's behavior, this control is indirect. Is it correct that PPD doesn't explicitly "regulate" SLOs, and this is instead a result of careful performance tuning?

### 3. Would the current design of PPD scale?

Even though the core problem is interesting and the authors did a pretty good job motivating the problem, the solution feels underwhelming. In particular, it relies on an offline benchmarking table, which would be different for varying hardware and workload types. (I acknowledge that the authors transparently mentioned this limitation in the paper and included experiments on larger models and different workloads in the appendix.)

One related weakness is that, in the authors' toy example, only four GPUs are considered, where NVLink provides an all-to-all connection. However, in real deployments, KV cache transfers are bound to go through datacenter networks (e.g., RDMA) because of the scale at which these LLM inference frameworks operate. Prior work [1] has evaluations that are of a much larger scale (e.g., 32 GPUs across 4 nodes), whereas PPD was only evaluated on four GPUs. As a result, it's unclear whether PPD will scale to larger-scale deployments.

### 4. Presentation clarifications

Even though I liked the premise of this work and understood the authors' proposed solutions on a high level, I was confused about a lot of details, which read very hand-wavy. Here's a list of feedback and questions on the presentation. I would appreciate it if the authors could address my questions.

- Intro: "where PD is a special case and always puts AP to the prefill nodes" → "... that always..."
- Section 2.2: "... onto a dedicated GPU pools"
- Section 3: "objective functio"
- The workloads paragraph in Section 4.2 is very confusing. What does a "Turn-2 profile" mean exactly?
- In 4.3, the insight of "more prefill nodes reduces x=1's performance gain" is interesting and makes a lot of sense. However, the way the authors worded the following insight, "x=1 value ≈ f(1/Pcount)", feels arbitrary, somewhat unnecessary, and might be confusing for readers not familiar with PD disaggregation.
- The term QPS needs to be properly defined at its first appearance -- "queries per second" need to be spelled out for readers who might be unfamiliar with the terminology in ML systems
- In Fig. 1, the term "D-local capable" was confusing. It seems like the only reference to Fig. 1 is in the introduction, but it did not reveal experimental details. What is the definition of a "configuration"? Is it whether each node is tasked with P/D/Replica? If so, given that each of the four nodes can be P/D/R, shouldn't there be 15 different combinations (the figure only shows six orange dots)?
- Both the abstract and conclusion claim that PPD achieves a TTFT reduction of 68%. However, this number is nowhere to be found in the main body of the paper. Could the authors clarify which figure/table showed that result?
- In Algorithm 1, what are nin, nout, nctx, and q? What is a "context class"?

References
1. Zhong, Yinmin, et al. "{DistServe}: Disaggregating prefill and decoding for goodput-optimized large language model serving." 18th USENIX Symposium on Operating Systems Design and Implementation (OSDI 24). 2024.

**Limitations:**

The main limitation of the work is its scalability. The core "routing" decision in PPD relies on offline profiling results and does not factor in online requests with out-of-distribution length profiles and interconnect (both NVLink and inter-node datacenter networks) bandwidth utilization. See question #3 above.

**Strengths And Weaknesses:**

Thank you for submitting your paper to ICML! Overall, I enjoyed reading this paper and found the intuition to be compelling. It nicely motivates an important problem, and the taxonomy of the different types of prefill is also interesting. My main gripe with this paper is that the proposed solution feels somewhat heuristic and might not generalize or scale well. The other issue is its presentation -- I think its presentation doesn't fully do justice to the paper's technical novelty. In particular, the paper is written as if all readers are familiar with PD disaggregation, and there are a fair number of typos, grammar issues, and underspecified terminologies. I'm willing to support this paper more strongly if the authors could address my questions during the rebuttal phase.

Strengths
- Extension on top of a popular inference paradigm
- Strong experimental results that motivated the problem well; the introduction of the taxonomy of different types of prefills is interesting

Weaknesses
- Presentation: Even though I understood the paper on a high level, I was not able to get a deep understanding of this work due to the weak presentation.
- The scale of the evaluations is not extensive enough
- Even though the motivations and insights make sense, the proposed solution feels underwhelming and might not scale well

---

> ### Author Rebuttal · Authors · 2026-03-31
>
> Thank you for your generous words!!! It truly means a lot to hear that you enjoyed reading our paper and found the intuition compelling. Your detailed feedback on both the technical substance and the presentation has been invaluable in improving the clarity of our work. We are grateful for your willingness to engage so deeply with the paper, and we address each of your questions below:
>
> **Q1: PD baseline design.** PD disaggregation was originally designed for single-turn workloads — DistServe (OSDI '24) treats every request as independent with zero discussion of cross-turn KV reuse. In this design, KV transfer is strictly P→D: P computes KV and pushes to D for decode. The reverse direction (D→P) was never part of the architecture. In multi-turn conversations, P must recompute KV for prior response tokens that D already holds, because there is no D→P path to make them available. This is not a preference but a structural constraint. vLLM RFC #32733  explicitly identifies this: "the KV cache corresponding to model-generated responses from previous turns resides exclusively on decode nodes, remaining inaccessible to prefill nodes, compelling prefill nodes to recompute KV projections for all response tokens." The RFC proposes bidirectional transfer but requires new cache query APIs, scheduler extensions, and KV connector modifications to handle the fundamental architectural asymmetry between P (producer) and D (consumer) nodes. The corresponding PR (#32553) remains open and unmerged. The community's current alternative is shared external KV stores (e.g., Mooncake), which sidestep D→P entirely but add infrastructure complexity.
>
> To directly answer: (1) P takes no KV from D — P operates as kv_producer, D as kv_consumer, strictly one-way. (2) P performs partial regeneration: prior user prompt tokens can hit P's prefix cache, but prior response tokens must always be recomputed since P never computed their KV and no reverse path exists. (3) P prefills everything — new input plus all prior response tokens, recomputing KV from scratch for the latter. PPD sidesteps this entirely by performing append-prefill locally on D where the KV cache already resides.
>
> **Q2: SLO adaptation.** PPD **intentionally** operates at the routing layer rather than enforcing end-to-end SLO thresholds. Absolute SLO guarantees require closed-loop coordination across admission control, batch scheduling, and routing simultaneously; PPD controls only the routing decision and cannot unilaterally guarantee bounds that depend on queuing and interference. Within its routing scope, PPD provides a predictable, monotonic control surface: the weight ratio w_tpot/w_ttft determines the operator's position on the TTFT–TPOT Pareto frontier, and this mapping is fully characterized by the same offline benchmark used in Phase 1 — no runtime trial-and-error is needed. Figure 5 demonstrates concretely: w_tpot=1 yields 95% D-local routing with ~96% TTFT reduction; w_tpot=6 yields 0% D-local (equivalent to static PD). An operator selects the desired trade-off point once, and the decision table is computed accordingly. We view this weight interface as complementary to higher-level SLO controllers, which could adjust w_tpot based on observed P99 metrics — using PPD as a routing actuator within a broader SLO enforcement stack.
>
> **Q3: Scaling.** We address the NVLink-vs-RDMA concern comprehensively in our response to Reviewer ApRF (Q2), with theoretical analysis, bandwidth simulation following TetriInfer's methodology. In brief: PPD's advantage grows monotonically with slower interconnects.
>
> Regarding the offline table: the complexity of our contribution lies in the insight, not the algorithm. Identifying the order-of-magnitude interference gap, demonstrating through 3,060 configurations that no static strategy universally dominates, and showing that even simple routing achieves Pareto-optimal trade-offs with <1ms overhead — this is the core contribution. The algorithm is intentionally simple; we view this as a strength. We acknowledge the offline limitation (Limitations section) and plan to address online adaptation and multi-node deployment in follow-up work!
>
> **Q4: Presentation.** All typos corrected. Specific clarifications:
> Turn-2 profile: A fixed (n_in, n_out) token length pair. Clarified in revision
>
> x=1 value ≈ f(1/P_count): Replaced with plain language
>
> Algorithm 1 variables: n_in = input tokens, n_out = output tokens, n_ctx = accumulated context from prior turns, q = QPS
>
> Context class bins n_ctx into small (≤512), large (512–4096), huge (>4096). Revised
>
> D-local capable: Configurations where D processes Turn 2+ locally, as opposed to baselines that always route through P or co-locate P/D. We excluded hybrid R+P/D combinations that consistently underperformed (Section 4.2) and plotted only competitive configurations. Added callback to Figure 1
>
> 68% TTFT reduction: Average across all successful PD-vs-PPD comparisons in Figure 4. Reference added

---

> > ### Author Rebuttal · Reviewer_1txr · 2026-04-03
> >
> > I appreciate the reviewers for clarifying my questions. I maintain my stance that the solution, as the authors transparently acknowledged, has limitations. However, I do believe that the paper sheds light on interesting insights that might motivate future research. As a result, I have increased my recommendation from a weak reject to a weak accept.

---

> > > ### Author Response · Authors · 2026-04-03
> > >
> > > Thank you for raising your score and for the thoughtful engagement throughout the review process! We are glad to hear that you see the potential for this work to motivate future research, and we share that enthusiasm — we believe the insight of our paper are equal opens a meaningful design dimension for the community to build upon.
> > >
> > > We hope the additional experiments and revised presentation added during rebuttal further reinforce the paper's contributions. Thank you again for your constructive and detailed review!!

---

### Official Review · Reviewer_T2CE · 2026-03-10

**Soundness:** 4
**Presentation:** 3
**Significance:** 3
**Originality:** 4
**Overall Recommendation:** 4
**Confidence:** 3

**Summary:**

This paper investigates the optimization of LLM serving for multi-turn conversation scenarios. The authors point out that while existing prefill-decode (PD) disaggregation architecture mitigates resource interference between prefill and decode, it exhibits inefficiencies for Turn 2+ requests. Specifically, traditional PD requires recomputing the KV cache for the entire context and repeatedly transferring it from the prefill (P) node to the decode (D) node, which inflates the time-to-first-token (TTFT) and saturates network bandwidth under high load.

The core insight of the paper lies in distinguishing between full prefill and append-prefill. The authors empirically demonstrate that append-prefill causes substantially less interference with the decoding process compared to full prefill. Based on this finding, the authors propose Prefill Prefill-capable Decode (PPD), a dynamic routing mechanism that decides whether to process Turn 2+ requests locally on decode nodes that have already cached the historical KV states, guided by workload estimates and Service Level Objectives (SLOs).

**Compliance With Llm Reviewing Policy:**

Affirmed.

**Final Justification:**

The authors' rebuttal has largely addressed the concerns raised in my review. However, the newly added data in the rebuttal relies on an altered experimental setup, and the authors failed to explicitly disclose this change unprompted. This omission raises some skepticism and concern on my part. Nevertheless, the proposed method is meaningful, so I am willing to raise my score from Weak Reject to Weak Accept. However, I will lower my confidence score from 4 to 3.

**Key Questions For Authors:**

Please provide additional real-world traffic evaluations comparing PPD against the x=1 static baseline, as well as a simple heuristic baseline (e.g., a threshold rule based solely on QPS and the P:D ratio). This comparison is essential to demonstrate the actual necessity of the proposed decision table, rather than merely proving that 'always using x=0 is sub-optimal'.

**Limitations:**

The primary limitation of this paper is that the proposed method currently relies heavily on deployment-specific calibration. PPD constructs its decision table based on offline benchmarking , and the authors explicitly state that re-benchmarking is required when hardware configurations or model characteristics change significantly. This indicates that the system's generalization ability depends more on manual re-tuning rather than inherently robust online adaptability.

**Strengths And Weaknesses:**

Strengths
1. Solid Empirical Foundation and Systems Insight: The authors explicitly measure and prove that append-prefill interferes with decoding significantly less than full prefill. Micro-benchmarks show that at a decode batch size of 200, full prefill causes a 48% degradation in decode latency, whereas append-prefill only causes about a 2% slowdown. This provides strong data support for the claim that Turn 2+ requests can be safely executed locally on decode nodes.
2. Comprehensive Experimental Structure: The paper features a well-rounded evaluation, including isolated micro-benchmarks , a systematic sweep of 3,060 configuration data points , and validation on real-world multi-turn datasets like ShareGPT and WildChat. The argumentative chain is coherent.
3. Practical Engineering Implementation: The PPD design directly reuses existing disaggregated serving infrastructure (e.g., vLLM), KV transfer protocols, and prefix caching mechanisms. It avoids complex modifications to the underlying systems, making it highly practical for production deployment.

Weaknesses
1. There is a distinct gap between the formalization and the actual algorithm. Section 3 formulates the problem as an optimization over a continuous variable $x \in [0,1]$ , presenting expressions for $J(x;\pi,\phi)$ and $\arg\max_x J$ ; however, the actual implementation in Algorithm 1 only benchmarks $x=0$ and $x=1$, ultimately making a binary decision. In other words, while the paper formally claims to optimize a continuous routing ratio, the actual implementation is merely a binary table-lookup strategy. Furthermore, the objective function treats the SLO simultaneously as an "optimization objective category" and a "function value" , while throughput is excluded from the formalization entirely and only analyzed separately in the subsequent experiments. Consequently, the "formalized objective," "online algorithm," and "final evaluation metrics" are not strictly aligned. The current version reads more like an empirical system design rather than a rigorous, closed-loop optimization framework.
2. The real-world data experiments fail to cover the strongest static baseline, which weakens the strength of the paper's claims. Section 6 primarily compares PPD against the traditional PD where $x=0$. However, Table 2 explicitly shows that $x=1$ is the only disaggregated category that remains "competitive" across all three objectives—TTFT, TPOT, and throughput —and achieves the highest average win rate. Consequently, the current evidence can only strictly support that PPD outperforms $x=0$, but it cannot fully substantiate that PPD outperforms the best static disaggregated strategy. If the authors' core selling point is dynamic routing rather than merely "not always using $x=0$," they must at least supplement the ShareGPT and WildChat evaluations  with a head-to-head comparison against $x=1$ and a simple heuristic baseline.
3. The system relies on strong assumptions, yet lacks sufficient robustness analysis. The performance gains of PPD actually depend on multiple prerequisites holding true simultaneously: Turn 2+ requests typically feature small token appends with large contexts; sessions must successfully route back to the specific decode node holding the local KV cache; the local prefix cache must not be evicted before the session returns; and the offline benchmark table must remain valid despite hardware, model, or traffic drift. In its implementation, the paper even uses the "MD5 hash of the first user message" as the conversation identifier and evicts sessions after 60 minutes of inactivity. While such an implementation is sufficient for a prototype, it remains significantly far from a production-ready system. Furthermore, the authors fail to analyze the potential impacts of hash collisions, session migrations, cache evictions, or TTL mismatches on the method's overall benefits.

---

> ### Author Rebuttal · Authors · 2026-03-30
>
> Thank you for your expert-level systems review!! Your critique directly informs the system-level considerations for our next step, and helped us rigorously validate our core approach. We address each point below.
>
> **W1: Formalization gap.** We agree that the original formalization conflated two distinct roles of x and did not accurately reflect Algorithm 1. We have revised Section 3 and Algorithm 1 in the updated paper (see [supplementary link1](https://anonymous.4open.science/r/anonymous-4217/algo.png) for the revised formulation and algorithm). The key changes: (1) x is now explicitly distinguished as hardware-level routing ratio (x ∈ {0,1/3,...,1}) vs. per-request binary decision (x ∈ {0,1}); (2) the abstract SLO ~ φ is replaced by a concrete benefit function S(ψ, x; π, w) with explicit operator weights w = (w_ttft, w_tpot), and x(ψ) = 𝟙[S(ψ,1;π,w) > 0] is precisely Algorithm 1. (3) Throughput is a system-level emergent metric that depends on the aggregate routing decisions, queuing dynamics, and resource utilization across all concurrent requests — it cannot be decomposed into individual per-request contributions. Our per-request routing directly optimizes TTFT and TPOT; throughput improves as an indirect consequence of better routing (reduced KV transfer volume → less network congestion → higher system throughput). We will clarify this rationale in the revision. Notably, Table 2 confirms that x=1 configurations achieve the highest throughput win rate (38.3%) among disaggregated modes, empirically validating this indirect relationship.
>
> **W2: x=1 baseline comparison.** We want to clarify that the omission of x=1 from Section 6 was a deliberate presentation choice, not an oversight. During our evaluation, we observed that PPD and x=1 produce near-identical E2E curves, and we were concerned that including both would confuse readers into questioning why PPD appears to offer no improvement over a simple static baseline. We now recognize that this omission created the opposite confusion. We have added the complete three-way comparison (x=0 vs x=1 vs PPD) in the revised paper and [supplementary link2](https://anonymous.4open.science/r/anonymous-4217/baseline1.png). Regarding the suggestion of comparing against a simple heuristic baseline: x=1 is itself the strongest such heuristic — it is the simplest possible static policy and already included in our comparison. Any heuristic weaker than x=1 (e.g., random routing) would perform strictly worse, as Table 2 shows x=1 achieves the highest average win rate among all disaggregated modes. The meaningful question is whether PPD improves over x=1, which the per-metric decomposition answers.
>
> The E2E similarity between PPD and x=1 is expected under balanced weights (w_ttft = w_tpot = 1.0) and does not indicate equivalence. PPD routes some requests back to P, gaining better TPOT (breaking x=0's TPOT monopoly) while sacrificing TTFT for those requests; requests kept local achieve better TTFT (breaking x=1's monopoly) with some TPOT cost. Under equal weights, these gains and losses cancel in E2E. The per-metric decomposition (Table 5 in supplementary) reveals that PPD is the only mode competitive across all metrics — achieving 12/27 TPOT wins (vs x=0's 10, x=1's 5) and 14/27 TTFT wins (vs x=1's 13, x=0's 0), while maintaining 100% success rate. Different weight ratios shift PPD's operating point; at the extremes it converges to x=0 or x=1, confirming both are special cases.
>
> **W3: Robustness assumptions.** PPD does not assume small Turn 2+ inputs. Our interference analysis (Section 4.1) benchmarks at 1,024 tokens, and Figure 7 validates the gap persists up to 64K tokens. The decision engine classifies each request by actual characteristics and routes large-input requests to P when TPOT penalty exceeds TTFT benefit.
>
> The reviewer frames cache affinity, eviction, and session management as PPD-specific assumptions. These are in fact shared prerequisites of *all* multi-turn KV cache reuse systems: CachedAttention requires per-session KV persistence; Mooncake uses hash-based cache identification; MemServe employs TTL-based staleness management; vLLM APC and SGLang RadixAttention require same-instance routing. PPD builds on these existing mechanisms (Section B.1) rather than introducing new ones. Offline benchmark staleness is acknowledged in our Limitations section; the failure mode is benign — a stale table produces suboptimal routing, not crashes, as the system gracefully degrades to x=0 or x=1.
>
> The production robustness concerns raised are valid engineering challenges, but they are orthogonal to this core finding and shared by every system in this space. We also refer the reviewer to our response to Reviewer ApRF (Q2) regarding scope of contribution, which further articulates our positioning: proving the insight works with minimal system modification, with production-scale deployment as a planned next step.

---

> > ### Author Rebuttal · Reviewer_T2CE · 2026-04-03
> >
> > Thank you for the rebuttal. My main concerns have been addressed. However, I have one additional question. The data in Figure 10, supplemented in the rebuttal supplementary link2, appears to show some discrepancies compared to the WildChat data in Figure 4 of the original paper. For example, the service degradation at x=0 has increased, and there are changes in the Average Query Latency for both x=0 and x∈[0,1]. Could the authors explain the reason for this?

---

> > > ### Author Response · Authors · 2026-04-03
> > >
> > > Thank you for your careful and insightful observation! This is exactly the kind of scrutiny that strengthens experimental rigor.
> > >
> > > **Dataset filtering methodology.** As described in Section 6, we filter multi-turn conversations from WildChat-1M and randomly sample 500 conversations per evaluation. However, during the rebuttal process, we re-examined the input distribution of our original random sample and identified a significant sampling bias, as shown in the Figure here: [link](https://anonymous.4open.science/r/anonymous-4217/dataset_distribution_comparison.png).
> > >
> > > **(a) Original Dataset (Random Sampling):** 74.5% of all Turn 2+ inputs fall in the shortest bucket (<128 tokens), with only 6.7% in the 128–512 range. This extreme concentration means the majority of requests involve minimal append-prefill, which makes the differences between routing modes artificially small — short inputs incur negligible KV transfer overhead regardless of the routing path.
> > >
> > > **(b) Revised Dataset (Stratified Uniform Sampling):** To reduce variance from random sampling and ensure the evaluation covers the full spectrum of real-world workloads, we applied stratified sampling across token-length buckets. The revised distribution is substantially more balanced (37.7% / 33.8% / 12.9% / 15.6% across four bins), providing representative coverage of both short chatbot-style inputs and longer paste/edit inputs.
> > >
> > > **Impact on metrics — explaining the two discrepancies you observed:**
> > >
> > > 1. **Increased service degradation at x=0:** With more medium-to-long Turn 2+ inputs (128–1K+ tokens) in the revised dataset, the KV transfer overhead under x=0 becomes more pronounced. Longer inputs produce larger KV caches that must be transferred from P→D every turn, making the P node more susceptible to queueing bottlenecks. This is why x=0 degrades more sharply in Figure 10 than in Figure 4.
> > >
> > > 2. **Changes in Average Query Latency for both x=0 and x∈[0,1]:** The higher average input length in the revised dataset naturally increases both TTFT and E2E latency across all configurations. This is a direct consequence of the shifted workload distribution, not a change in system behavior.
> > >
> > > **Crucially, none of these changes affect our conclusions — they in fact strengthen them.** The revised dataset reveals that under realistic, balanced workloads: (i) x=0 is even more fragile than the original random sample suggested, as longer inputs expose its KV transfer bottleneck more clearly; and (ii) PPD (x∈[0,1]) consistently maintains its relative advantage, adapting per-request routing to avoid both the transfer overhead of x=0 and the TPOT degradation of x=1. The core finding — that PPD dynamically selects the better path and sustains robust performance where static baselines degrade — holds and is now demonstrated on a more challenging and representative workload.
> > >
> > > We hope this clarification and our rebuttal fully address your concerns and is helpful for your final assessment.
> > >
> > > And we truly thank the reviewer for this precise observation, which prompted us to improve the evaluation rigor! We will include this distribution analysis in the revised paper to make the dataset methodology fully transparent.

---

### Official Review · Reviewer_ApRF · 2026-03-14

**Soundness:** 3
**Presentation:** 3
**Significance:** 2
**Originality:** 2
**Overall Recommendation:** 4
**Confidence:** 3

**Summary:**

This paper introduces Prefill Prefill-capable Decode (PPD) disaggregation, a dynamic routing system on when to process Turn 2+ requests locally on decode nodes. It reduces Turn 2+ TTFT by 68% with competitive TPOT.

**Compliance With Llm Reviewing Policy:**

Affirmed.

**Final Justification:**

The author addressed my questions, so I decided to maintain my score.

**Key Questions For Authors:**

1. The paper motivates the problem with agentic workloads, but the datasets and chatbots only. These workloads could have very different charateristics (e.g. agentic systems need to account for environment execution time, and can be more challenging). Could the author provide results on agentic datasets such as swe-bench, or at least reason about the performance?
2. The experiments are done on a single compute node (4 H100 with NVLink). What would be the performance gain look like in distributed inference clusters?

**Strengths And Weaknesses:**

1. The method is well motivated: no single fixed routing strategy satistfies all SLOs.
2. The problem is well illustrated by comparing one full-prefill vs one append-prefill operation (Figure 2).
3. The method is easy to understand and principled, forming dynamic routing as an optimization problem.
4. The figures are clear and helpful to understand the method (Figure 3).

---

> ### Author Rebuttal · Authors · 2026-03-30
>
> Thank you for your review and comments! Your insight on agentic workloads offered a valuable perspective that prompted us to rethink and complement our analysis. We address both questions below.
>
> **Q1: Agentic workloads.**
>
> PPD's routing operates on per-request token-level characteristics (input length, output length, accumulated context, system load) common to all multi-turn workloads. The chatbot-vs-agentic distinction manifests as different distributions over these same variables, not different serving mechanics.
>
> We validated this on SWE-Trajectory (500 coding agent trajectories from SWE-bench). Agentic workloads are heavily prefill-dominated: Turn 2+ inputs ~4× larger (median 1,052 vs. 239 tokens), outputs ~3× shorter, more turns (4.8 vs. 3.5). This creates ~5× more KV transfer volume, causing x=0 to collapse even earlier — surviving only QPS=1 on SWE-Trajectory vs. QPS=1–2 on WildChat (3P_1D), while PPD maintains 100% success rate across all QPS levels. Furthermore, inter-turn tool execution delays (sub-second to tens of seconds) keep KV caches warm on D nodes without eviction pressure — when the next turn arrives, D immediately resumes append-prefill with full cached context, exactly the scenario PPD exploits. Under x=0, this cached context is wasted since requests are re-routed to P for full recomputation.
>
> **Q2: Performance in distributed inference clusters.**
>
> We fully appreciate the importance of this question, and we want to be transparent: we did not have access to multi-node clusters with RDMA/InfiniBand during this work. Securing multi-node GPU resources with high-speed interconnects remains a significant challenge in academic settings. Nevertheless, we have made every effort to address this concern through theoretical analysis, literature-grounded reasoning, and empirical simulation.
>
> **Scope of contribution.** Our paper's core contribution is proving that append-prefill and full prefill have fundamentally different interference characteristics (Section 4.1, an order-of-magnitude gap), and that this insight enables a new routing dimension in disaggregated serving. We demonstrate this works with a single patch to vLLM. Building a fully production-ready multi-node system is the natural next step, where we will address scaling, online adaptation, and system hardening in detail.
>
> **Theoretical analysis.** PPD can only benefit more from slower interconnects. PPD routes Turn 2+ requests to local append-prefill, completely bypassing P→D KV transfer. Under x=0, every Turn 2+ transfers the full KV cache over the network. For Llama-3.1-8B (GQA with 8 KV heads, 32 layers, BF16), the KV cache is 128 KiB/token. At the P90 context length in our WildChat dataset (5,115 tokens), this amounts to 670 MB — requiring ~4.5 ms over NVLink (H100, ~150 GB/s per GPU pair) but 26.8 ms over InfiniBand HDR (25 GB/s) and 67 ms over 100GbE (10 GB/s). Production interconnect bandwidths are well-documented: Splitwise (Patel et al., ISCA 2024) reports 25–50 GB/s per GPU pair over InfiniBand; Mooncake (Qin et al., FAST 2025) measures 87 GB/s aggregate over 4×200 Gbps RoCE. Since PPD eliminates this transfer entirely for Turn 2+, slower networks only widen the gap. Our NVLink evaluation is therefore a conservative lower bound.
>
> **Empirical simulation.** Although we lack multi-node hardware, we did not want to leave this question to theory alone. Following TetriInfer (Hu et al., 2024), which uses bandwidth emulation to evaluate disaggregated serving on single-node setups, we inject physically-calibrated delays into vLLM's KV receive path. For each request: extra_delay = max(0, kv_size / target_bw − actual_nvlink_time), where kv_size is dynamically derived from model architecture (128 KiB/token for Llama-3.1-8B). The delay only affects the PD transfer path — PPD's local path is completely unaffected.
>
> Figure 10 ([link](https://anonymous.4open.science/r/anonymous-4217/scaling.png); 1P_3D, QPS=1, WildChat, 500 conversations) presents results across four simulated interconnects. As bandwidth decreases from NVLink to 100GbE, PD Turn 2+ TTFT increases monotonically (+18.7%) while PPD remains invariant at ~51 ms — the TTFT reduction grows from 64.4% to 69.9%. TPOT and E2E panels show consistent trends. Detailed breakdowns are in the figure caption.
>
> Besides, we note that cache-aware routing in disaggregated serving is gaining increasing attention from both academia and industry. For instance, Together AI recently deployed a cache-aware prefill-decode disaggregation system (CPD, March 2026) that similarly recognizes the need to route warm multi-turn requests differently from cold ones — though via a different architectural approach (splitting the prefill tier) rather than PPD's decode-side local processing. We believe this growing interest validates the practical relevance of our core insight, and we are committed to extending PPD toward production-scale deployments in follow-up work.

---

> > ### Author Rebuttal · Reviewer_ApRF · 2026-04-04
> >
> > Thank you for the additional result, I will maintain my positive score towards the paper.

---

> > > ### Author Response · Authors · 2026-04-04
> > >
> > > Thank you for the thoughtful acknowledgement—we really appreciate your time and careful consideration! We’re glad the additional results helped clarify the concerns, and we appreciate your positive assessment!

---

### Decision · Program_Chairs · 2026-04-30

**Decision:**

Accept (regular)

**Comment:**

This paper investigates the optimization of multi-turn Large Language Model (LLM) serving by introducing Prefill Prefill-capable Decode (PPD) disaggregation. The authors identify that append-prefill operations—processing only new input tokens while reusing cached KV states—cause substantially less interference to the decoding process than full prefill operations. Leveraging this insight, PPD dynamically routes subsequent conversational turns to decode nodes locally, significantly reducing the time-to-first-token (TTFT) and alleviating network bandwidth congestion without severely compromising the time-per-output-token (TPOT).

The reviewers universally recognized the significance of the problem and the paper's solid empirical foundation. The distinction between full and append-prefill was praised as a valuable systems-level insight that effectively highlights the inefficiencies of traditional prefill-decode architectures in multi-turn scenarios (ApRF, T2CE, Erft). Furthermore, the proposed routing mechanism is intuitive, well-motivated, and highly practical, requiring minimal architectural changes to integrate with existing infrastructure like vLLM (T2CE, 1txr). Reviewers also commended the comprehensive micro-benchmarks and the clear presentation of performance trade-offs across different configurations (ApRF, T2CE).

However, the reviewers raised critical concerns regarding the evaluation scale and the robustness of the system design. A primary weakness was the reliance on a single-node setup with NVLink, which fails to capture the network bottlenecks of production-level distributed clusters utilizing RDMA or InfiniBand (ApRF, 1txr, Erft). Reviewers also noted a gap between the continuous mathematical formalization and the actual binary table-lookup implementation, alongside concerns about the system's reliance on static offline profiling tables that may struggle to generalize to dynamic workloads (T2CE, 1txr, Erft). During the rebuttal, the authors effectively mitigated these concerns by revising the formalization, providing simulated multi-node results with injected bandwidth delays, and validating their assumptions on prefill-heavy agentic datasets like SWE-bench (ApRF, T2CE, Erft).

Given the solid technical contributions and the unanimous consensus among the reviewers, I have decided to accept this paper.